# Potential-dependent polaron formation activates TiO$_2$ for the hydrogen evolution reaction

Tongwei Wu [1,10] ✉, Xiaoxi Guo[2,10], Guangjie Zhang[3], Yanning Zhang [1], Li Song [4], Zheng Liu [5] ✉, Hui Zhang [6], Shucheng Shi[7] ✉, Limin Liu [8], Marko M. Melander [9] ✉ & Karoliina Honkala [9] ✉

Polarons play a crucial role in determining the (photo)electrocatalytic activity of semiconductors. Traditionally, polarons are introduced ex situ and irreversibly during catalyst synthesis, but herein we present a fundamentally different approach of introducing polarons in situ in a reversible manner using the external electrode potential. We study the potential-dependent polaron formation and its impact on electrocatalysis on a prototypical TiO$_2$ semiconductor electrode for the acidic hydrogen evolution reaction. By combining grand canonical ensemble density functional theory calculations with (in situ spectro) electrochemical experiments, we demonstrate notable changes in TiO$_2$´s electronic structure driven by the reduction of Ti$^{4+}$ to Ti$^{3+}$ surface polarons at reducing potentials. Our results show that potential-dependent polaron formation creates highly active sites for the hydrogen evolution reaction, breaks down the linear relationship between adsorption energy and electrode potential, and leads to complex electrochemical reaction kinetics. We discuss how the in situ polaron generation can be leveraged in improving semiconductor (photo)electrodes. Overall, our findings provide compelling evidence and an atomistic understanding of potential-dependent polaron formation in semiconductor (photo)electrocatalysis.

Electrocatalysis and photoelectrocatalysis play a central role in facilitating clean energy conversion and enabling numerous sustainable processes and future technologies[1–8]. Realizing cost-effective, large-scale utilization of these technologies requires the development of (photo)electrocatalysts exhibiting high activity, selectivity, and stability[4–20]. While noble metals are the most common and widely studied electrocatalysts, recent focus has shifted toward noble metal-free semiconductor electrocatalysts, such as metal oxides/sulfides[5,21–26]. Although native semiconductors typically exhibit poor electrical conductivity due to a band gap, modifying their electronic properties through structural defects (e.g., anion vacancies, cation vacancies, and lattice distortions), phase transitions, conductive

[1]Institute of Fundamental and Frontier Sciences, University of Electronic Science and Technology of China, Chengdu, China. [2]Precision Medicine Translational Research Center, West China Hospital, Sichuan University, Chengdu, China. [3]CAS Key Laboratory of Standardization and Measurement for Nanotechnology, National Center for Nanoscience and Technology, Beijing, China. [4]National Synchrotron Radiation Laboratory, CAS Center for Excellence in Nanoscience, University of Science and Technology of China, Hefei, China. [5]School of Materials Science and Engineering, Nanyang Technological University, Singapore, Singapore. [6]Shanghai Synchrotron Radiation Facility, Shanghai Advanced Research Institute, Chinese Academy of Sciences, Shanghai, China. [7]Center for Transformative Science, ShanghaiTech University, Shanghai, China. [8]School of Physics, Beihang University, Beijing, China. [9]Department of Chemistry, Nanoscience Center, University of Jyväskylä, Jyväskylä, Finland. [10]These authors contributed equally: Tongwei Wu, Xiaoxi Guo. ✉e-mail: twwu77@uestc.edu.cn; z.liu@ntu.edu.sg; shishch1@shanghaitech.edu.cn; marko.m.melander@jyu.fi; karoliina.honkala@jyu.fi

additives, and surface strain offers pathways to enhance both conductivity and electrocatalytic activity[11,17]. Nevertheless, atomic-scale insight into how defects and electronic properties impact semiconductor electrocatalytic activity under operation conditions remains elusive despite extensive investigations.

Currently, atomic-scale understanding of semiconductor electrocatalysis typically relies on the use of descriptors that link the electronic or geometric structure of a semiconductor with its electrocatalytic activity[10,12,14,16,27]. These descriptors are commonly obtained through atomistic simulations, particularly using density functional theory (DFT). Consequently, several theoretical concepts, including the $e_g$ orbital occupancy[27], band-gap energy model[28,29], and binding energy-based scaling relations[30,31], have been developed to establish correlations between the semiconductor active site (electronic) structure and electrocatalytic activity. However, these descriptors are usually derived from calculations that do not explicitly consider the electrode potential. As a result, they fail to capture the potential-dependent behavior at the electrode-electrolyte interfaces[32–34]. They also do not accurately represent the effect of charge accumulation or depletion, or the formation of localized electronic states, known as polarons[32–36]. Since polarons are closely associated with potential-dependent changes in both geometry and electronic structure[7], traditional constant charge DFT approaches cannot capture these intricate effects influencing semiconductor electrocatalysis[4,6].

Polarons and defects have also been associated with (photo) electrocatalytic activity[4,11,23,36], and recent experimental studies have highlighted a strong correlation between semiconductor electrocatalytic activity and polaron formation. For example, a computational study[37] has shown that the presence of $Ce^{3+}$ polarons in the $Pt_1$/$CeO_2$ catalyst leads to an increased electrocatalytic activity towards oxygen reduction reaction (ORR). By increasing the $Ce^{3+}$ polaron concentration, the ORR overpotential of $Pt_1$/$CeO_2$ decreases from 0.47 V to 0.16 V as the polarons act as electron traps and thereby reduce the charge transfer from reaction intermediates to radicals. An experimental study demonstrated band edge shifts in ultrathin ZnO films as a function of the potential, resulting in the formation of new (polaronic) electronic states and changes in charge transfer kinetics[38]. These and other experimental results[33,35,36] explicitly demonstrate that the electronic structure, polaron formation, and related electrocatalytic activity of semiconductors can efficiently be controlled through the application of an electrode potential. Hence, it is essential to include electrode potential-driven changes in the geometric and electronic structure in computational studies of semiconductor electrochemistry.

Besides computational and electrochemical experiments, insight into changes in the charge distribution and electronic structure under operation conditions can be obtained through in situ/*operando* spectroscopic techniques such as in situ X-ray photoelectron spectroscopy (XPS), X-ray absorption near-edge structure spectroscopy (XANES), electron paramagnetic resonance (EPR) spectroscopy, and ultraviolet-visible (UV-Vis) spectroscopy[39–47]. For example, a $Ni(OH)_2$ monolayer was probed with operando XANES to demonstrate that the $Ni^{2+}$ state can be oxidized to $Ni^{3+}$ and $Ni^{4+}$ at more positive potentials[48]. These changes further promote oxygen vacancy ($V_O$) formation in the $Ni(OH)_2$ monolayer and markedly improve the oxygen evolution (OER) activity[48]. Recently, microcell-based in situ electronic/electrochemical measurements for a $MoS_2$ semiconductor electrode demonstrated that the conductivity of the $MoS_2$ surface enhances with increasing surface charge concentration, and the enhanced conductivity is responsible for high electrocatalytic hydrogen evolution (HER) activity[35]. On $Fe_2O_3$ hematite, high photoelectrochemical water splitting activity arises from its capability to form multiple hole polarons on the electrode surface under light illumination and the application of oxidizing electrode potential[49].

While all these in situ studies highlight that the electrode potential can qualitatively change the electronic structure, control polaron formation, and quantitatively impact the catalytic activity of semiconductor electrodes, the state of a catalyst, the exact nature of active sites, and reaction mechanisms under operation conditions remain debated[15,24,42,50]. Resolving these questions requires detailed atomic-scale insight into the potential-dependent changes in the semiconductor (electronic) structure and their influence on electrocatalytic performance. For example, the activity of acidic HER has been identified to correlate with the presence of defects and polarons[26,33,35,36,51]. For 2H-$MoS_2$, it has been observed that sulfur (S) vacancies can improve the HER performance by introducing polaron gap states near the Fermi level[52]. The HER performance has also been linked to oxygen vacancy density and subsequent polaron formation in $TiO_{2-x}$[53]. However, the interplay between the electrode potential and polaron/defect formation, and their correlation with HER activity, remains an unresolved question.

In this study, we investigated how electrode potential activates the prototypical semiconductor (photo)electrode anatase $TiO_2$ for HER. As a widely studied model (photo)electrode, $TiO_2$ is known to be a highly stable, non-toxic, and low-cost material[54], and it can achieve high (photo)electrocatalytic HER activity when doped with other metals (see Supplementary Table 9). We utilized state-of-the-art constant inner potential (CIP-) DFT[6] calculations to explore how the potential-dependent polaron formation influences the thermodynamics and kinetics of HER on anatase $TiO_2$. Our computational predictions were verified through various (spectro)electrochemical experiments, providing clear evidence on the potential-dependent polaron formation and its pivotal role in enhancing HER activity. By analyzing the electronic structure and activity of $TiO_2$ catalysts, we unravel the precise nature of active sites and the reaction mechanisms under operating conditions. Finally, we propose strategies to enhance the activity of semiconductor electrocatalysts by controlling polaron formation.

## Results and discussion
### Potential-dependent polaron formation mechanisms on TiO₂ (101)

We used CIP-DFT[6] as implemented in the GPAW[55,56] software to study the most stable[54] anatase $TiO_2$ (101) surface. All calculations were spin-polarized, utilizing the DFT + U approach with a self-consistently computed $U_d$ (Ti) value of 4.5 eV for the $TiO_2$ system[54,57]. $TiO_2$ (101) exhibits both 5- and 6-fold coordinated Ti ($Ti_{5c}$ and $Ti_{6c}$) sites along with 2-fold coordinated bridging oxygen ($O_{br}$) and 3-fold coordinated oxygen ($O_{3c}$) sites (see Supplementary Fig. 2a). CIP-DFT calculations on an ideal, uncharged $TiO_2$ (101) surface correspond to the conditions of potential of zero charge (PZC), achieved at a potential of 1.0 V vs. SHE ($V_{SHE}$). A projected density of states (PDOS) plot in Fig. 1a shows that Ti atoms have a spin-paired electron configuration as the atomic magnetic moments (AMM) of Ti are zero, and as the DOS does not display spin-dependency (see Supplementary Information Section S1 for using DOS to analyze polaron formation). The vacant 3 $d$ orbitals of Ti atoms dominate the conduction band. These features indicate that Ti cations of a neutral $TiO_2$ (101) surface are in the quadrivalent state ($Ti^{4+}$–3$d^0$)[58–60] and that no electron polarons exist.

$Ti^{3+}$ polaron formation and its electrode potential dependency are demonstrated in Fig. 1b–e. Figure 1b (orange line) shows that at 0 $V_{SHE}$ the $TiO_2$ (101) surface carries a negative charge of 0.29 e$^-$ while the PDOS plot (Fig. 1c) displays that the $E_f$ moves up in energy and is located at the bottom of the conduction band. As no spin polarization is seen in the PDOS, the excess charges are not localized on any particular atom, and thus no electron polarons are formed in Ti-3$d$ orbitals. Further decreasing electrode potential raises $E_f$, leading to the accumulation of electrons at the surface. The PDOSs plots (Fig. 1d, e) at −1.0 $V_{SHE}$ and −2.0 $V_{SHE}$ show that the initially unoccupied Ti-3$d$

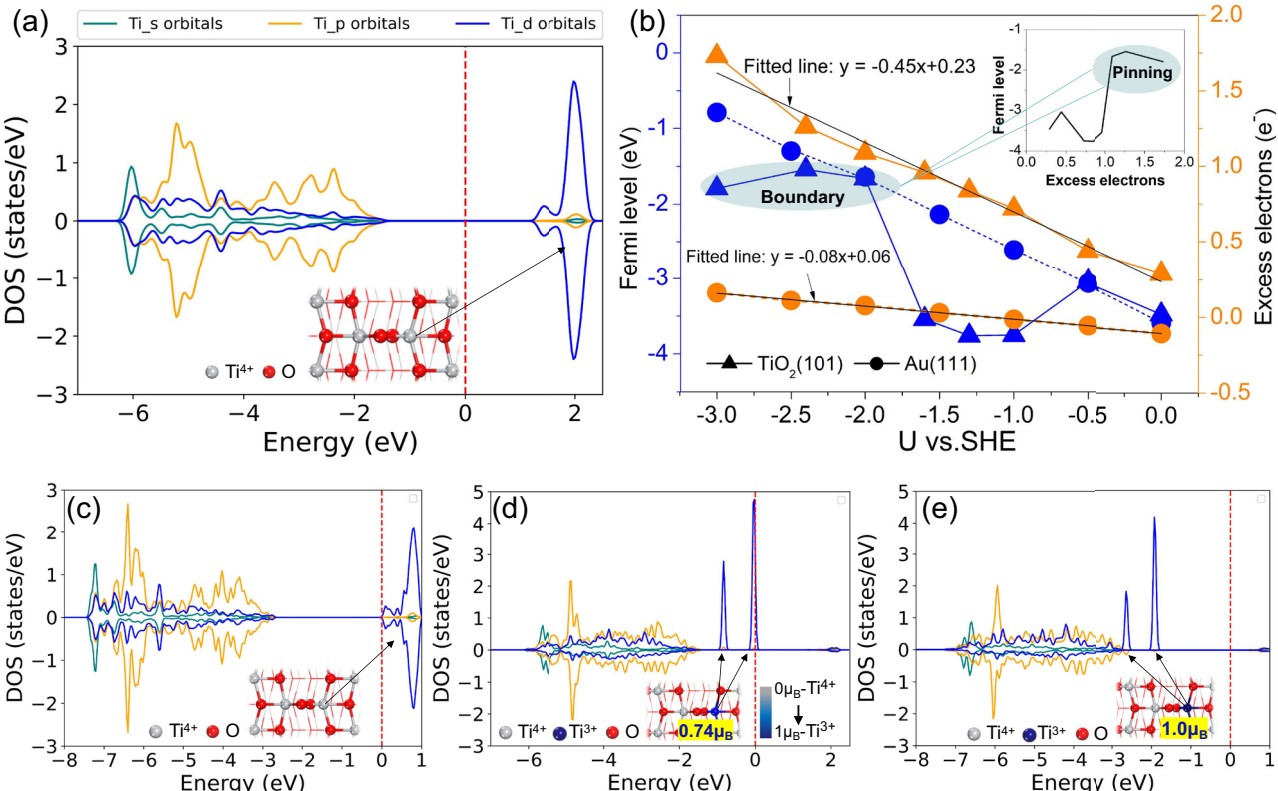

**Fig. 1 | The charge environment under constant potential conditions. a** The electronic and atomic structures of the $TiO_2$ (101) surface at PZC of 1.0 $V_{SHE}$. The red dashed line corresponds to the position of the Fermi level, $E_f$. The color code in DOS plots: green for *s* electrons, yellow for *p* electrons, and blue for *d* electrons of Ti atoms. **b** The variations of $E_f$ and excess electrons as a function of electrode potential for $TiO_2$ (101) (diamonds) and Au (111) (circles) surfaces. The black lines in electron accumulation correspond to a linear fit to the data, and negative numbers

mean a positively charged surface. The inset figure presents the Fermi level as a function of the number of excess electrons. The DOS analyses of the electronic structure and polaron formation of the $TiO_2$ (101) surface at **c** 0 $V_{SHE}$, **d** −1.0 $V_{SHE}$ and **e** −2.0 $V_{SHE}$. Atomic structures show the top view of the $TiO_2$ (101) surface. Ti cations are grey, O anions are red, and blue stands for the $Ti^{3+}$ cation with a polaron state.

orbitals of a $Ti_{6c}$ surface atom are shifted below $E_f$. This shift is accompanied by the splitting of a single DOS peak into two peaks (Fig. 1d). In particular, the DOS peak emerging below the Fermi level at U = −1.0 $V_{SHE}$ is due to the localization of excess electrons (0.72 $e^-$) and a considerable AMM of 0.74 $\mu_B$ on a single Ti atom (Fig. 1b and 1d): this indicates the formation of a localized electron polaron state at U = −1.0 $V_{SHE}$. Figure 1d further demonstrates that the presence of these states depends sensitively on the electrode potential. The application of a highly reducing potential, U = −2.0 $V_{SHE}$, leads to further electron accumulation (1.09 $e^-$) and shifts the initially unoccupied Ti-3*d* orbitals completely below the Fermi level. Since the AMM of the Ti cations is ~1.0 $\mu_B$, we conclude that the $Ti^{4+}$ is fully reduced to $Ti^{3+}$ with a clear polaronic state (Fig. 1e). Decreasing the electrode potential to U = −3.0 $V_{SHE}$ leads to accumulation of more electrons (1.73 $e^-$) on the surface. These excess electrons localize on $Ti_{5c}$ and $Ti_{6c}$ surface atoms, with AMMs of 1.0 $\mu_B$ and 0.75 $\mu_B$ (Supplementary Fig. 5), respectively, indicating the formation of a bipolaron state at U = −3.0 $V_{SHE}$[61].

The CIP-DFT simulations allow us to understand the potential-dependent polaron formation and surface charging. With decreasing the electrode potential from 0 $V_{SHE}$ to −3.0 $V_{SHE}$, the negative surface charge of $TiO_2$ (101) increases linearly. The slope in Fig. 1b corresponds to the electronic capacitance and shows that the $TiO_2$ (101) capacitance (orange diamonds) is about six times smaller than that of a typical metallic electrode, Au (111) (circles in Fig. 1b). Figure 1b also demonstrates that the $E_f$ of $TiO_2$ (101) does not depend linearly on the potential or charge. Instead, the Fermi level of $TiO_2$ (101) is pinned at an electrode potential of c.a. −2.0 $V_{RHE}$ and remains almost constant even when changes in the electronic structure (polaron formation) take

place[62,63]. We attribute the pinning to the large number of occupied surface states (polarons) at the valence band under highly negative electrode potentials. The $E_f$ pinning may have an important role in semiconductor electrocatalysis, as noted in the recent experimental work for the hematite electrode[64] where the formation of electron polarons in the reduction of Fe from $Fe^{3+}$ to $Fe^{2+}$ pins the Fermi level and impacts charge transfer rates.

The potential-dependent electron paramagnetic resonance (EPR) spectroelectrochemical results in Fig. 2a provide direct experimental evidence for the formation of $Ti^{3+}$ polaron states under highly reducing conditions. The EPR signal (signal center at B = 355 mT) for $Ti^{3+}$ polaron formation is absent at potentials higher than U = −2.1 $V_{SHE}$. The signal first appears at U = −2.1 $V_{SHE}$ and increases as the electrode potential becomes more negative. Notably, the $Ti^{3+}$ polaron formation potentials from in situ electrochemical EPR measurements and CIP-DFT calculations (Figs. 1e and 2a) show good agreement, with a potential deviation of only 0.1 $V_{SHE}$. To conclusively demonstrate potential-dependent polaron formation, we conducted in situ near-ambient pressure X-ray photoelectron spectroscopy measurements (NAPXPS) as a function of applied electrode potential, as shown in Fig. 2b and Supplementary Fig. 36. Upon applying a reducing potential of U = −2.2 $V_{SHE}$, a new XPS peak emerged at lower binding energy, which we attribute to the formation of $Ti^{3+}$ polaron on the $TiO_2$ surface. This onset potential for $Ti^{3+}$ polaron formation determined in situ NAPXPS agrees closely (within 0.1 $V_{SHE}$) with both in situ EPR data and CIP-DFT simulations. Given the surface-sensitivity of in situ NAPXPS, we conclude that potential-dependent formation of $Ti^{3+}$ polarons mainly occurs on the $TiO_2$ surface, which is consistent with our computational results.

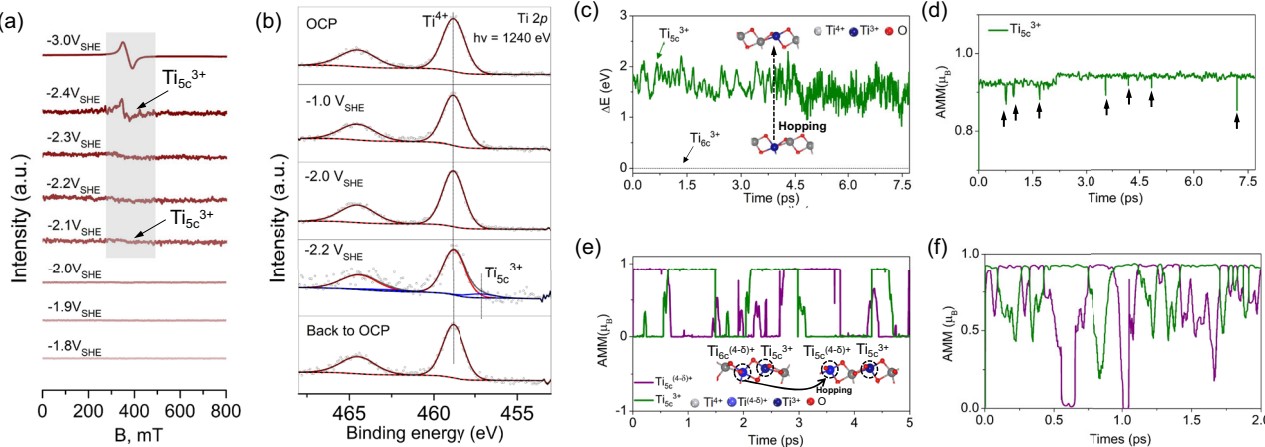

**Fig. 2 | The dynamic behavior of potential-dependent Ti3+ polarons.**
**a** In situ EPR spectroelectrochemical measurements of a TiO$_2$-modified ITO electrode at different potentials in 0.1 M H$_2$SO$_4$ electrolyte (without iR-correction, pH = 1.0). **b** In situ NAPXPS spectroelectrochemical measurements of a TiO$_2$ electrode at different potentials in 0.1 M H$_2$SO$_4$ electrolyte (without iR-correction, pH = 1.0). **c** The dynamic behavior of the Ti$^{3+}$ polaron state on TiO$_2$ (101) is calculated with the CIP-DFT-MD. The dashed line represents the energy of Ti$_{6c}^{3+}$ polaron calculated

with the static CIP-DFT approach at U = −2.0 V$_{SHE}$. **d** The variation of AMM for a Ti$_{5c}^{3+}$ polaron state on TiO$_2$ (101) at U = −2.0 V$_{SHE}$. The arrows indicate times where the AMM value exhibits the most notable fluctuations. **e** The variation in AMM for Ti$_{5c}^{3+}$/Ti$_{5c}^{(4-\delta)+}$ polaron states on TiO$_2$ (101) at U = −2.4 V$_{SHE}$. δ in the superscript indicates the number of excess electrons (0 <δ ≤ 1). **f** The variation of AMM for Ti$_{5c}^{3+}$/Ti$_{5c}^{(4-\delta)+}$ polaron states on TiO$_2$ (101) at U = −3.0 V$_{SHE}$. Ti cations are grey, O anions are red, and blue stands for the Ti$^{3+}$ cation with a polaron state.

To verify that the polaron formation is reversible and not caused by irreversible structural changes, we performed control experiments (Fig. 2b) showing that the Ti$^{3+}$ signal disappears when the potential is returned to the open-circuit potential (OCP), where no polaron signal was observed. The reversibility of polaron formation was further confirmed by in situ EFM (Supplementary Fig. 10), EPR (Supplementary Fig. 11), and Raman spectroscopy (Supplementary Fig. 34) measurements. Together, these experimental results establish that the potential-driven Ti$^{3+}$ polaron formation is a fully reversible in situ process rather than a permanent structural transformation.

To further elucidate the stability and dynamics of the Ti$^{3+}$ polarons at room temperature at varying potentials, we performed CIP-DFT molecular dynamics (MD) simulations[6,65–69]. While CIP-DFT-MD provides the possibility of directly simulating potential-dependent polaron formation and dynamics in semiconductors[4,6], it should be noted that achieving the experimental (milli)second regime, e.g., NAPXPS or EPR measurements, is not possible with (CIP-)DFT, machine learning potential, or even classical MD simulations. Nevertheless, the CIP-DFT-MD simulation in Figs. 2, 3, and Supplementary Fig. 13 complements the experiments and directly demonstrates that potential-driven Ti$^{3+}$ surface polarons are both thermodynamically and dynamically stable at room temperature under reducing conditions on the picosecond timescale. In particular, Fig. 2c shows that a Ti$^{3+}$ polaron localizes on the Ti$_{6c}$ site upon structure optimization at U = −2.0 V$_{SHE}$. However, in the CIP-DFT-MD simulations, the polaron instantaneously moves from a Ti$_{6c}$ site to a Ti$_{5c}$ surface site due to thermal motion and this change is accompanied by a small energy penalty (see the energy difference (ΔE) in Fig. 2c), which is counterbalanced by an entropy gain from the availability to more polaronic states. Monitoring the AMM of Ti$_{5c}^{3+}$ over time shows that the magnetic moment remains stable and exhibits only minor thermal fluctuations, as indicated by black arrows in Fig. 2d and Supplementary Fig. 12.

At a more reducing potential of U = −2.4 V$_{SHE}$, a bipolaron can intermittently localize either on two neighboring Ti atoms or the next-nearest neighboring Ti atoms with frequent transitions between these configurations, see Fig. 2e. By further decreasing the electrode potential to U = −3.0 V$_{SHE}$ the bipolaron state shows higher electron occupancy, which correlates with an increased oscillation frequency of AMM values in the bipolaron state (Fig. 2f). The large oscillations in the AMM values suggest that a bipolaron state may be unstable at room

temperature and that its stability further decreases with increasing electron occupancy.

The computational and experimental results in Figs. 1 and 2 clearly show that the TiO$_2$ (101) can store a high amount of negative charge by reducing Ti$^{4+}$ to Ti$^{3+}$ polarons when the electrode potential is sufficiently reduced. The polarons are formed and observed exclusively at the top layer of TiO$_2$ (101) surface, which is expected to enhance surface conductivity, see Supplementary Fig. 9. The formation of surface polarons is in line with previous simulations[15,18], in situ microcell-based electronic/electrochemical[35], and STM[53] experiments demonstrating that charge carriers localize and conduct charge at the semiconductor surfaces rather than in the bulk under negative electrode potential conditions.

## The effect of an oxygen vacancy on polaron formation in TiO$_2$ (101)

TiO$_2$ often contains oxygen vacancies and other point defects, which may play a decisive role in its (photo)electrocatalytic performance[53,58–60]. While TiO$_2$ can host various defects, the O$_{br}$ (bridge-site oxygen) is generally regarded as the energetically most favorable surface oxygen defect on anatase TiO$_2$ (101)[54,58]. Therefore, we considered the impact of the O$_{br}$ vacancy on the polaron generation. The V$_O$ formation decreases the coordination numbers of Ti$_{6c}$ and Ti$_{5c}$ atoms to Ti$_{5c}$ and Ti$_{4c}$, and leaves behind two excess unpaired electrons, which can lead to the formation of two Ti$^{3+}$ ions, i.e. a Ti$^{3+}$ bipolaron (bi-Ti$^{3+}$) state[61,70] (see Fig. 3a at PZC condition).

The V$_O$ formation energies were calculated at various potentials, and a linear correlation between the electrode potential and defect formation energy is observed (Supplementary Table 9). The results also show that the accumulation of negative surface charge (Ti$^{3+}$ polarons) is driven by the applied electrode potential, and that V$_O$ shifts the polaron formation potentials to significantly more oxidizing potentials compared to pristine TiO$_2$. This implies that oxygen vacancies facilitate electron uptake and the formation of electron polarons. Furthermore, the charge and spin analyses confirm that the bi-Ti$^{3+}$ is localized next to V$_O$ at PZC of 0.5 V$_{SHE}$, as seen by the non-zero AMMs of two Ti atoms (Ti$_{5c}$ and Ti$_{4c}$) neighboring the vacancy (Fig. 3a) and occupied Ti-3$d$ states (see Fig. 3b). The PDOS plot (Fig. 3c) of the Ti atom demonstrates that one Ti$^{3+}$ in bi-Ti$^{3+}$ has partially unoccupied Ti-3$d$ states and indicates the presence of a single-polaron state at 1.5

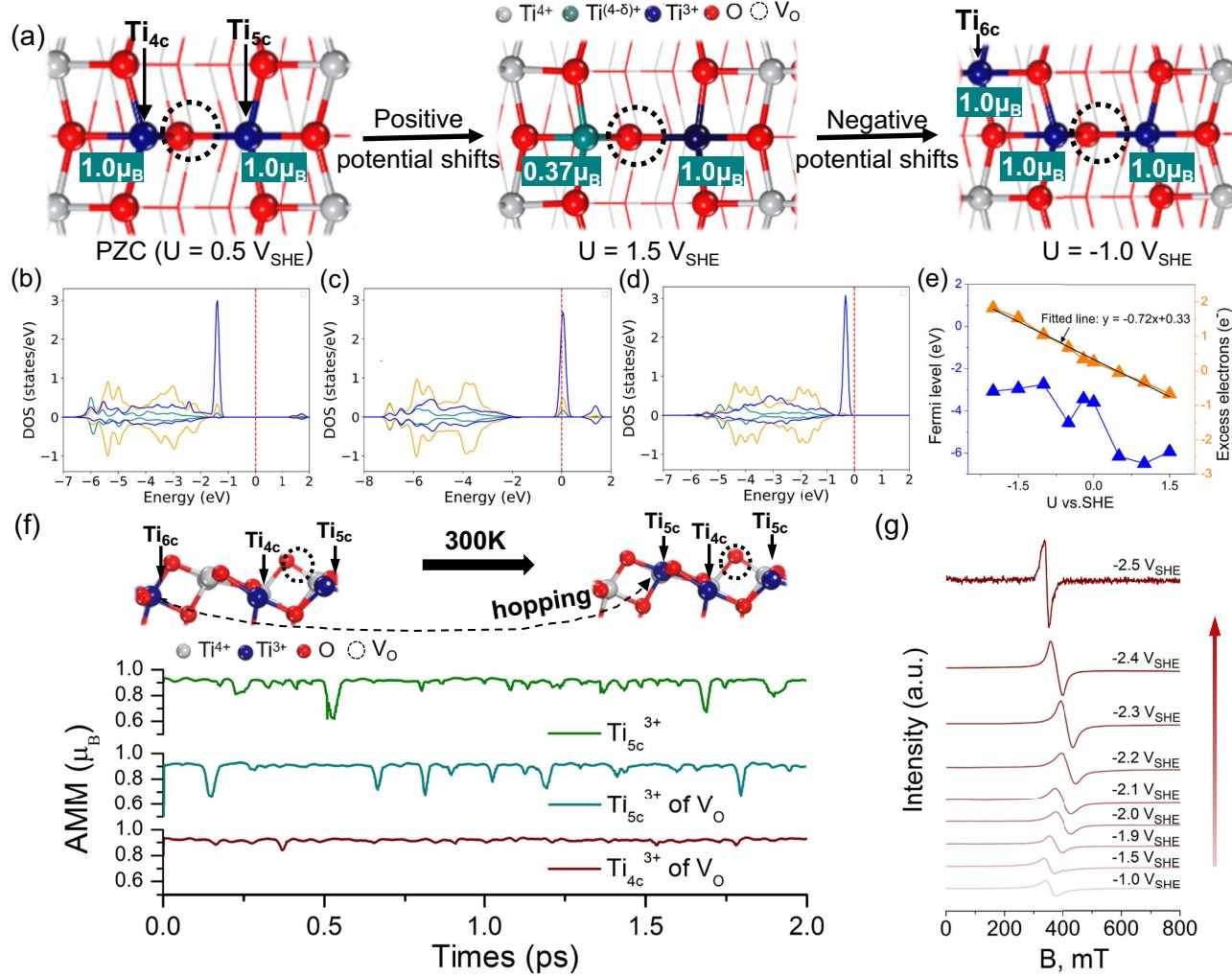

**Fig. 3 | The properties of potential-dependent Ti$^{3+}$ polarons on the oxygen-defected TiO$_2$ surface. a** The top view of the defected surface. The dashed black circle represents a missing O$_{br}$ atom. **b** DOS plot of the Ti$_{4c}$ atom with 1 μ$_B$ at PZC of 0.5 V$_{SHE}$. **c** DOS plot of Ti$_{4c}$ atom with 0.37 μ$_B$ at U = 1.5 V$_{SHE}$. **d** DOS plot of additional Ti$_{6c}$ atom with 1.0 μ$_B$ at U = −1.0 V$_{SHE}$. **e** The variation of E$_f$ and excess electrons as a function of electrode potential for V$_O$-TiO$_2$ (101). **f** The dynamic behavior of Ti$^{3+}$ polaron state on the V$_O$-TiO$_2$ (101) surface calculated with the CIP-DFT-MD at

U = −1.0 V$_{SHE}$ and corresponding changes in the AMM of Ti$^{3+}$ polarons. The dashed arrow represents the hopping pathway of the Ti$^{3+}$ polaron. **g** In situ EPR spectro-electrochemistry of a V$_O$-TiO$_2$-modified conducting glass (ITO) electrode at different potentials in 0.1 M H$_2$SO$_4$ electrolyte (without iR-correction, pH = 1.0) recorded in a spectroelectrochemical setup. Ti cations are grey, O anions are red, green stands for the Ti$^{3+}$ cation with a partially localized polaron state, and blue stands for the Ti$^{3+}$ cation with a fully localized polaron state.

V$_{SHE}$. At this potential, the surface carries a positive charge of 0.66e$^-$ (Fig. 3e), with only one Ti$^{3+}$ as evidenced by the AMM values shown in Fig. 3a. Decreasing the electrode potential to −1.0 V$_{SHE}$ leads to the accumulation of an additional electron and the formation of a second or even a third Ti$^{3+}$ polaron with AMM of 1.0 μ$_B$ (see Fig. 3a and 3d).

Figures 1b and 3e also demonstrate a more oscillatory relationship between E$_f$ and electrode potential on the V$_O$-TiO$_2$ (101) than on the pristine TiO$_2$ (101) surface. We tentatively attribute these larger oscillations to the V$_O$-facilitated electron accumulation and polaron formation. Furthermore, we examined the stability and dynamics of multiple Ti$^{3+}$ polarons in V$_O$-TiO$_2$ using CIP-DFT-MD simulations at U = −1.0 V$_{SHE}$. Changes in the AMM and polaron locations in Fig. 3f show that a complete bi-Ti$^{3+}$ state is trapped by oxygen vacancy[71], while the potential-induced third Ti$^{3+}$ polaron (Ti$_{6c}$) hops to a Ti$_{5c}$ surface site, forming a continuous Ti$^{3+}$ polaron chain. This thermal polaron hopping has a very small energy penalty (Supplementary Fig. 13), which indicates that potential-induced Ti$^{3+}$ polarons in V$_O$-TiO$_2$ exhibit enhanced conductivity due to facilitated polaron hopping as compared to pristine TiO$_2$. These computational results are supported by the experimental in situ EPR spectroelectrochemical and

electrochemical impedance spectroscopy (EIS) results in Fig. 3g and Supplementary Fig. 14 for a defective TiO$_2$ surface. The EPR signal for a Ti$^{3+}$ polaron state exhibits a more drastic increase on the V$_O$-TiO$_2$ than observed on pristine TiO$_2$ at a given electrode potential, which is attributed to the improved surface conductivity[70,72] resulting from the enhanced charge accumulation in TiO$_2$ and facilitated polaronic charge hopping between Ti$^{4+}$ and Ti$^{3+}$ [73,74].

## Experimental results for the potential-dependent polaron formation on electrocatalytic activity

The impact of polarons on electrocatalytic activity was examined for the acid HER as a prototypical electrocatalytic reaction. The linear sweep voltammetry (LSV) experiments reveal that a V$_O$-TiO$_2$ nanoarray electrode requires an overpotential of −0.35 V$_{SHE}$ to achieve a HER current density of 20 mA cm$^{-2}$ (Fig. 4a). In contrast, the pristine TiO$_2$ demands a substantially more reducing potential, −0.70 V$_{SHE}$, to attain the same current density. At a higher overpotential of −1.5 V$_{SHE}$, the V$_O$-TiO$_2$ nanoarray electrode reaches a current density of 260 mA cm$^{-2}$, which is significantly higher than the value measured on pristine TiO$_2$ nanoarray (37 mA cm$^{-2}$). To shed light on the structural stability of

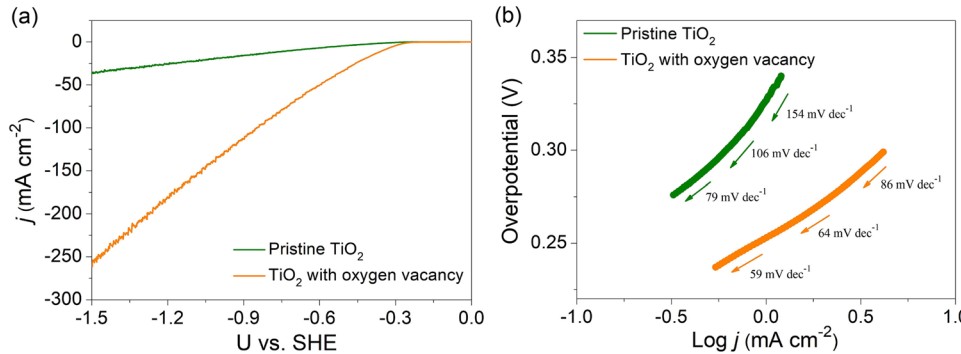

**Fig. 4 | HER performance. a** The current polarization curves of HER on both pristine $TiO_2$ and oxygen vacancy modified-$TiO_2$ in 0.1 M $H_2SO_4$ electrolyte (without iR-correction, pH = 1.0). **b** Tafel plots of both pristine $TiO_2$ and oxygen vacancy modified-$TiO_2$.

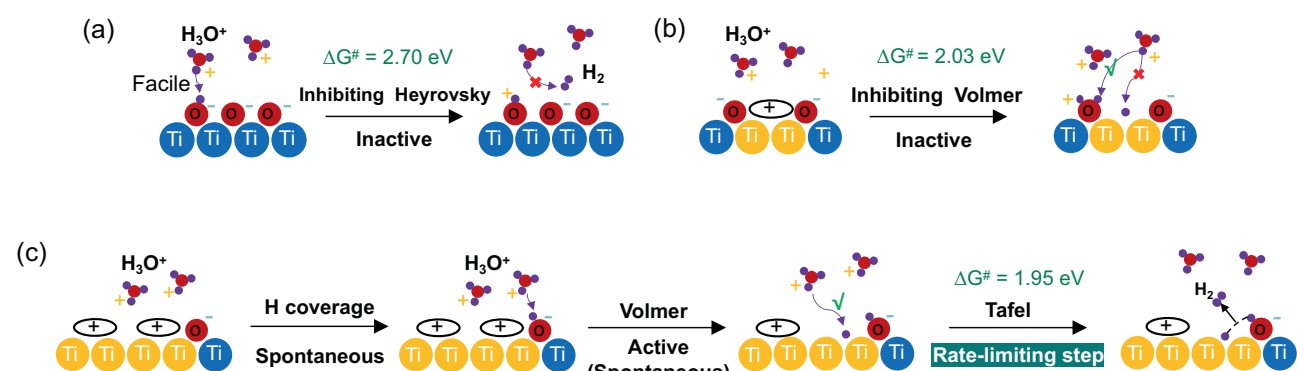

**Fig. 5 | Schematic illustration of the HER mechanisms.** HER mechanisms on **a** pristine $TiO_2$ at U = 0 $V_{SHE}$, **b** $V_O$-$TiO_2$ at U = 0 $V_{SHE}$, and **c** $V_{2O}$-$TiO_2$ at U = 0 $V_{SHE}$, close to the experimental onset potential. $Ti^{4+}$: blue; $Ti^{3+}$: yellow; O: red; H: purple. The "+" represents a buildup of more positive surface charge induced by the removal of a surface $O_{br}$ atom and the formation of the OH group. The "-" represents a buildup of more negative surface charge induced by the surface $O_{br}^{2-}$ atom.

pristine and defective $TiO_2$ electrodes, we also utilized in situ Raman spectroscopy. Supplementary Fig. 33 shows that the Raman signals are significantly diminished under highly reducing potentials. As discussed in the Supplementary Information Section S7, the Raman spectra indicate notable changes in the Ti-O structure and its Raman response, which we ascribe to potential-dependent $Ti^{3+}$ polaron formation[75,76]. Furthermore, the weakened $E_g$, $B_{1g}$ and $A_{1g}$ characteristic peaks in $TiO_2$, which are induced by potential-dependent $Ti^{3+}$ polaron formation, return to their initial state under open-circuit conditions after removal of the electrode potential at U = −3.0 $V_{SHE}$, see Supplementary Fig. 34. It suggests that the structure changes of $TiO_2$ induced by potential-dependent $Ti^{3+}$ polaron formation are a reversible process. Overall, the low HER activity of the pristine $TiO_2$ is attributed to the absence of polaronic states, as evidenced by the computational results as well as EPR spectra discussed earlier. Conversely, the higher HER activity of the $V_O$-$TiO_2$ electrode is ascribed to the presence of oxygen vacancies and polarons.

To further explore the role of vacancies and polarons on the acidic HER activity, we conducted a Tafel analysis for both electrodes. Figure 4b shows that the Tafel slopes of a $V_O$-$TiO_2$ nanoarray depend slightly on the electrode potential, ranging between 59 and 86 mV/dec. On pristine $TiO_2$, the spread of Tafel slopes exhibit broader range, spanning from 79 to 154 mV/dec depending on the electrode potential. These Tafel slopes strongly indicate that on pristine $TiO_2$ HER follows the Volmer-Heyrovsky mechanism with the Heyrovsky step as the rate-determining step[77]. While a single Tafel slope alone cannot fully quantify the intrinsic electrocatalytic activity of either material, comparison of the Tafel slopes reveals that the intrinsic HER activity of $V_O$-$TiO_2$ nanoarray is significantly higher than that of pristine $TiO_2$. We also

note that the Tafel slopes of $V_O$-$TiO_2$ at lower overpotentials are close to the cardinal value of 60 mV/dec, commonly considered as an indicator of HER following the Volmer-Tafel mechanism, with the Tafel step identified as the rate-determining step (see Supplementary Information Section S10)[77,78].

## Computational analysis on the impact of polarons and oxygen vacancies on HER activity

To understand the atomic-scale mechanisms of HER on the $TiO_2$ (101) surface, we simulated the acidic HER thermodynamics and kinetics at different electrode potentials. On the pristine surface, the Volmer reaction ($H_3O^+ + e^- \rightarrow H^* + H_2O$)[79–81], is very facile and cannot be the rate-determining step as shown in Fig. 5a. We therefore addressed the following Heyrovsky step ($H^* + H_3O^+ + e^- = H_2 + H_2O$), which was found to have very high reaction barriers on pristine $TiO_2$ (101) even under highly reducing conditions where HER is experimentally seen to take place. Consequently, the Heyrovsky step appears kinetically slow and limits HER activity on the ideal surface due to the strongly bound surface proton[82]. These findings are in line with our experimental results and we conclude that on pristine $TiO_2$ HER follows that Volmer-Heyrovsky mechanism, where the latter step is rate-limiting. We didn't further consider the Volmer-Tafel pathway because the formation of two strongly bound hydrogens is expected to result in a very large barrier for the Tafel step. Overall, we find that the pristine $TiO_2$ (101) surface is very inefficient towards HER, as the hydrogen adsorption on this surface is very strong.

In the presence of a single oxygen vacancy ($V_O$-$TiO_2$ forming two $Ti^{3+}$ polarons), the Volmer step at the $V_O$-site has very high activation energies at U = 0 $V_{SHE}$ (Table 1), which limits the HER activity on the

defected surface. This is inconsistent with the experimental observation where the Tafel step is identified as the rate-determining step. Therefore, we considered the impact of a double oxygen vacancy ($V_{2O}$-$TiO_2$, forming four $Ti^{3+}$ polarons, Supplementary Fig. 21b) on the Volmer step, focusing on the potential range between $U = 0$ $V_{SHE}$ and $U = -0.3$ $V_{SHE}$. In this step, a hydrogen atom prefers to adsorb on the oxygen atom nearest to the oxygen vacancy, leading to the formation of an OH group, see Fig. 5c and Supplementary Fig. 20a. At this stage, another hydrogen atom spontaneously adsorbs at the $V_O$ site, bridging two neighboring $Ti^{3+}$ atoms through a second Volmer step. As the Volmer kinetics on $V_{2O}$-$TiO_2$ appear feasible, we considered the nominally chemical Tafel step ($2H^* \rightarrow H_2$). Our calculations show that the Tafel step is kinetically hindered at $U = 0$ $V_{SHE}$ (Fig. 5c) and the activation energy decreases when the potential decreases to $U = -0.3$ $V_{SHE}$ (see Table 1). Thus, the Tafel step appears to be the rate-determining step on the $V_{2O}$-$TiO_2$ (101) as it has the highest activation energy. The experimental Tafel slope analysis further supports this conclusion, as we find that HER follows the Volmer-Tafel mechanism with the Tafel

step as the rate-determining step. Overall, we conclude that $V_{2O}$-$TiO_2$ is the most likely active surface for the HER, where HER follows the Volmer-Tafel mechanism with the Tafel step as the RDS.

## Impact of potential-dependent polaron formation on free energy relations, kinetics, and electrocatalyst design

Our simulations and experiments show that the electronic structure, surface charge, and reaction energetics depend non-trivially on the electrode potential due to potential-dependent polaron formation (see Supplementary Figs. 22–24 and related discussion). To gain deeper insight, we carried out a detailed analysis of the potential-dependent adsorption energies and the linear Brønsted-Evans-Polanyi (BEP) relationship, which links activity with reaction energy. This approach, commonly employed for metal electrodes, helps to extrapolate measurements at low over-potentials to predict electrocatalytic performance under the relevant reaction conditions.

The results in Fig. 6a show that on $TiO_2$(101), the linear CHE-like relationship between hydrogen adsorption energy and electrode potential breaks down within the potential range where polaron formation takes place. While linear or slightly quadratic free energy relationships between the hydrogen adsorption and electrode potential hold for a wide range of metallic electrodes[33,83], the observed breakdown suggests that the potential-dependent activity of semiconductor electrodes differs fundamentally from that of metals. Therefore, the commonly used linear relationships appear unsuitable for semiconductor electrodes, and the presence of polaronic states suggests the need for caution when evaluating thermodynamics as an explicit function of potential.

We also calculated the transfer coefficient of the Volmer step and the Tafel steps on the $V_{2O}$-$TiO_2$ surface using the formula $\Delta G^{\#} = \Delta G_0 - \alpha eU$, where $\alpha$ is the transfer coefficient. We found that for the Volmer step, $\alpha = 1.42$ and $1.45$ for the first and second Volmer step (Supplementary Fig. 25a). For the Tafel step, $\alpha = 1.22$, see Supplementary

**Table 1 | Grand canonical reaction free energies ($\Delta G_r$) and activation free energies ($\Delta G^{\#}$) for the elementary reaction steps of HER on pristine $TiO_2$ (101) and $V_O$/$V_{2O}$-site on $V_O$/$V_{2O}$-$TiO_2$ (101) surfaces**

| Surface | Reaction step | U ($V_{SHE}$) | $\Delta G_r$ (eV) | $\Delta G^{\#}$ (eV) |
|---|---|---|---|---|
| Pristine $TiO_2$ | Volmer | 0.0 | Spontaneous | 0.0 |
| | Heyrovsky | 0.0 | 1.75 | 3.12 |
| | | −0.3 | 1.20 | 2.70 |
| $V_O$-$TiO_2$ | Volmer | 0.0 | 0.50 | 2.03 |
| $V_{2O}$-$TiO_2$ | Volmer | 0.0 | Spontaneous | 0.0 |
| | Tafel | 0.0 | 0.63 | 1.95 |
| | | −0.3 | 0.22 | 1.52 |

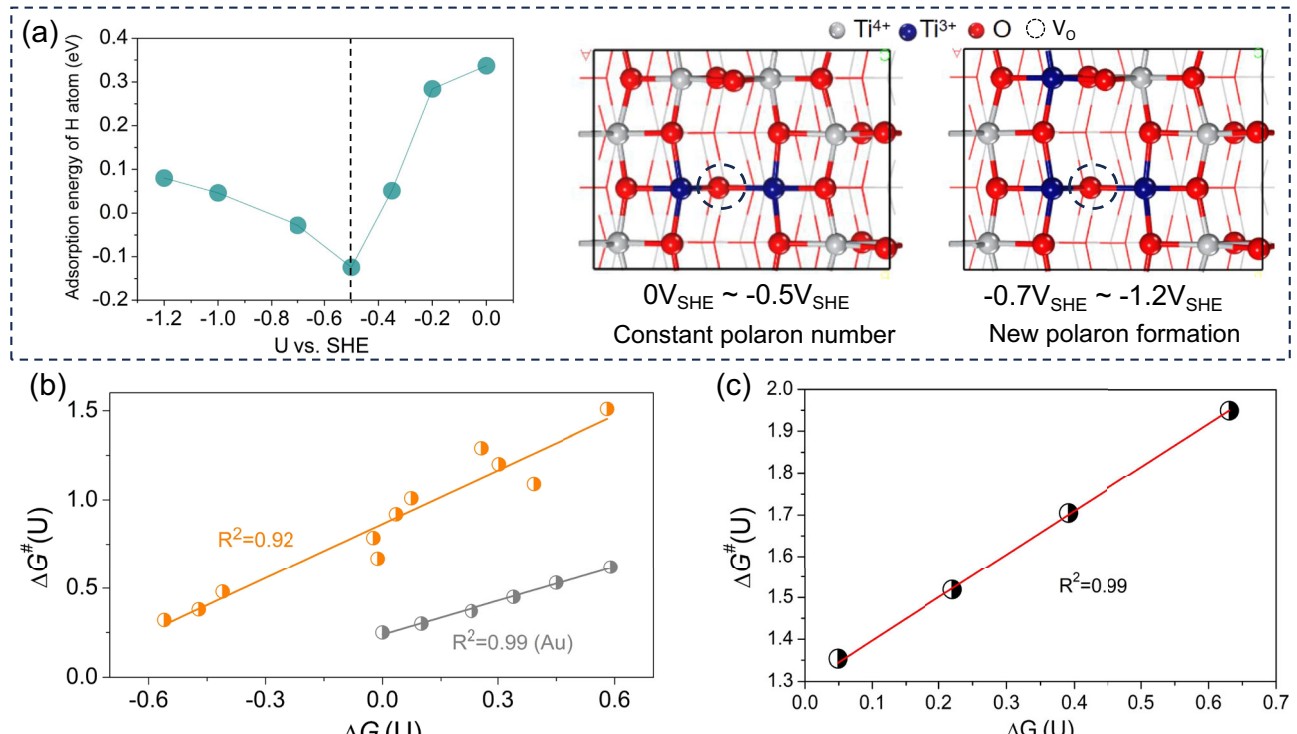

**Fig. 6 | The catalytic mechanism of potential-dependent $Ti^{3+}$ polarons. a** The relationship between hydrogen adsorption energy and the electrode potential, and a top view of the Vo-defected $TiO_2$ surface. **b** The BEP linear relationship between $\Delta G^{\#}$ and $\Delta G_r$ for the Volmer reaction as a function of electrode potential on $V_{2O}$-

$TiO_2$ and Au. **c** The BEP linear relationship between $\Delta G_r$ and $\Delta G^{\#}$ for the Tafel reaction as a function of electrode potential on $V_{2O}$-$TiO_2$. The $\Delta G^{\#}$ and $\Delta G_r$ values for Au are taken from literature [83]. Ti cations are grey, O anions are red, and blue stands for the $Ti^{3+}$ cation with a polaron state.

Fig. 25b. These results suggest that the electrode potential plays a crucial role in lowering the barrier for both the electrochemical Volmer step and the nominally chemical Tafel step. Meanwhile, based on $b = (2.3RT)/(\alpha F)$, we obtained a Tafel slope of 48.5 mV/dec, which is close to the experimental value of 60 mV/dec. However, it should be noted that in the Butler-Volmer model $\alpha$ should be between zero and one, and our observation that $\alpha$ is larger than one suggests that the limitations of the Butler-Volmer theory might not always hold for semiconductor electrodes (see Supplementary Information Section S10). Here, the underlying reason for $\alpha$ to be larger than one is the concomitant and potential-dependent polaron formation and electron transfer in HER steps. This suggests that future research on semiconductor electrocatalysis might require revising the Butler-Volmer model.

However, the BEP relationship between reaction and activation energies holds for both the Volmer (Fig. 6b) and the Tafel (Fig. 6c) steps on the TiO$_2$ electrodes. These findings suggest that the reaction barrier can be inferred by simulating potential-dependent reaction thermodynamics, see Fig. 6b, and thus, the BEP lines obtained at low overpotentials could be extrapolated to predict electrocatalytic performance under the relevant reaction conditions[84].

Overall, the potential-dependent polaron formation leads to complex dependencies between the reaction thermodynamics, kinetics, and applied potential. Because the linear relationship between the adsorption energy and the applied electrode potential is broken while the linear BEP correlation is still valid, one may still estimate the reaction barriers through BEP relationships after a careful analysis of the reaction energies and accounting for polaron formation. These insights imply that once the non-linear dependency between polaron formation, electrode potential, and reaction thermodynamics is understood, the in situ control of potential-dependent polaron formation offers a practical way to tune the activity of electrocatalytic reactions on semiconductor electrodes. These strategies are not limited to TiO$_2$, as we show in Supplementary Figs. 26–28 for NiO – other plausible materials include transition metal oxides, such as BiVO$_4$[36] and hematite[49] (Fe$_2$O$_3$), where polaronic states control (photo)electrocatalytic activity. However, a more in-depth investigation of catalytic processes in semiconductor electrodes is needed, as their electronic structure is more sensitive to reaction conditions, such as electrode potential, than that of metal electrodes.

We have combined our state-of-the-art constant potential methodology with various electrocatalytic experiments to show that the application of an electrode potential leads to the reduction of Ti$^{4+}$ atoms to Ti$^{3+}$ polarons at the TiO$_2$ surface. As we show that the polarons control the HER thermodynamics, kinetics, conductivity, and the overall HER activity, our work unveils a way to control semiconductor (photo)electrocatalysis through potential-driven polaron formation. Due to the significant non-linear changes in the surface charge and reaction energetics induced by polaron formation, revised methods are needed to understand, explain, and predict electrocatalytic performance when potential-dependent polaron formation takes place. In addition to offering fundamental insight into HER on semiconducting electrodes, we also suggest that leveraging the synergy between defects and potential-dependent polaron formation could serve as a strategy to design improved (photo)electrocatalysts.

## Methods
### Computational details
All simulations were carried out using density functional theory (DFT) as implemented in the GPAW software[55,56] version 19.8.1. The exchange-correlation effects were accounted for using the BEEF-vdW-functional, which combines the generalized gradient approximation with the Langreth-Lundqvist van der Waals-functional to achieve accurate adsorption energies[85]. A 3×3×1 k-point mesh was used, and all the calculations were spin-polarized. To ensure a robust description of the electronic structure Hubbard correction U$_d$ (Ti) = 4.5 eV was utilized.

The U value was obtained through the self-consistent linear response DFT + U method (LR-DFT + U)[57].

The constant potential, grand canonical ensemble DFT calculations were carried out using the constant inner potential (CIP) DFT method[6]. In CIP-DFT, the inner potential, i.e., the average electrostatic potential, of the electrode is fixed to a constant value, which enables explicit control over the electrode potential. The particular advantage of CIP-DFT over other GCE-DFT methods is the possibility to continuously control the electrode potential even across the band gap region and to model potential-dependent polaron formation, making CIP-DFT highly useful for simulating semiconductor electrochemistry[4]. The absolute electrode potential $U_M(abs)$ is computed from the electrode inner potential on the potential-of-zero-charge (PZC) scale, and is then determined as:

$$U_M(abs) = U^M(\sigma) - U^M(\sigma = 0) = \phi^M(\sigma) - \phi^M(\sigma = 0)$$

As shown in the Supplementary Information Section S1. Here, the electrode inner potentials, $\phi^M(\sigma)$, are referenced against the solution inner potential, $\phi^S(\sigma) = 0$, which is set to zero using Dirichlet boundary conditions in the Poisson equation from which the electrostatic potential is solved.

The absolute electrode potential vs the standard hydrogen electrode $U_{SHE}$ is further defined as below:

$$U_{SHE} = U^M(abs) - U^{SHE}(abs)$$

Where $U^{SHE}(abs)$ has been determined experimentally to be ~4.44 V[4]. The energy used in the analysis of electrode reactions is the grand free energy:

$$\Delta\Omega(\phi^M; N_e) = F(N) - \left(\mu^0 - \phi^{\Delta PZC}\right)N_e$$

The reaction grand free energy ($\Delta G_r$) for each elementary reaction was calculated as

$$\Delta G_r = \Delta\Omega$$

The dynamics of explicitly solvated electrochemical interfaces under constant potential conditions were performed using constant inner potential (CIP-DFT) molecular dynamics (MD)[6,66–69]. Langevin dynamics (friction = 0.2 ps$^{-1}$) was used for the TiO$_2$ (101) system to control the temperature (set at 300 K) in the simulations with 1 fs time step and 2 u mass for hydrogen atoms[86]. The trajectories were computed with the following convergence criteria: eigenstates: 1.0e$^{-4}$, density: 1.0e$^{-5}$, energy: 1e$^{-6}$ setting. Note that the potential-dependent TiO$_2$ (101) surfaces were thermalised for 2 ps before the production runs.

The solvent at the electrochemical interface was modeled using a hybrid of implicit/explicit approach combining four-layer explicit solvent comprising 16 H$_2$O molecules in TiO$_2$(101) and a two-layer explicit solvent comprising 10 H$_2$O molecules in Au (111), and a SCMVD[87] dielectric continuum model for water was exploited as implicit solvent to fill the rest. The atomic radii used by SCMVD to determine the cavity size were set to 2.2 Å for Ti and 1.5 Å for O. The positions and orientations of the explicit water molecules were optimized using the minima[88] hopping global optimization method as implemented in ASE[89].

The theoretical lattice constants for Au (4.24 Å) and for TiO$_2$ (a = b = 3.836 Å and c = 9.841 Å) were used, which were found by minimizing the total energies of bulk Au and TiO$_2$. The computational lattice constants are slightly overestimated compared to experimental values of Au (4.08 Å)[90] and TiO$_2$ (a = b = 3.782 Å and c = 9.502 Å)[91,92]. The TiO$_2$ (101) system was modeled as a 2 × 1 supercell with four (O-Ti-O) tri-layers with the bottom trilayer fixed. The Au (111) surface was modeled

as a 2 × 2 supercell with Au four layers with the bottom layer fixed. 25 Å of dielectric solvent was added on top of the slabs. The transition states of each elementary reaction step were found using the constrained minimization approach[93], and they were confirmed by the presence of a single imaginary vibrational mode along the reaction coordinate. A solvated $H_3O^+$ is considered as the initial state for the proton transfer reaction.

For the determination of $Ti^{3+}$ polaron formation, while the primary role of DOS is to quantitatively describe the distribution of electronic energy levels, integrating the DOS over the given energy range ($\int_{E_1}^{E_2} \rho(\epsilon)d\epsilon$) provides the number of states between the energies $E_1$ and $E_2$.

However, this approach does not provide quantitative information about the spatial distribution of electrons. By correlating the atomic spin magnetic moment and DOS, it is possible to effectively capture both the spatial localization and the energy level of e.g., a polaronic state. This method is particularly applicable to non-magnetic $TiO_2$ systems where the quantitative relationship between magnetic moment and electron localization is clear. This is due to the single source of magnetic moment and the absence of spin coupling interference. The atomic magnetic moments in $TiO_2$ arise from excess electrons forming polarons so the DOS and atom spin magnetic moments together quantitatively describe electron localization and energy.

**Chemicals.** Ti mesh (TM) was purchased from Shengshida Metal Production Co., Ltd. (80 mesh, Xingtai, China). Sodium hydroxide (NaOH, A. R. grade) was obtained from Aldrich Chemical Reagent Co., Ltd. (Shanghai, China). Hydrochloric acid (HCl, A. R. grade) and sulfuric acid ($H_2SO_4$, A.R. grade) were supplied by Sinopharm Chemical Reagent Co., Ltd. (Shanghai, China). All the reagents were used as received. The water used throughout all experiments was purchased from Wahaha Group Co., Ltd. (Hangzhou, China).

**Preparation of pristine-TiO₂.** The pristine-$TiO_2$ array was grown on the Ti mesh substrate via a hydrothermal method[94]. Initially, a clean TM piece (3 cm × 3 cm) was immersed in 40 mL 5 M NaOH for 1 hour. Afterwards, the TM and solution were all transferred to a 50 mL Teflon-lined stainless-steel autoclave and maintained at 200 °C for 20 h. After cooling the autoclave to room temperature, the prepared precursor was soaked in a 0.5 M HCl solution for 1 h to facilitate the exchange of $Na^+$ with $H^+$. Finally, the sample was thoroughly rinsed with deionized water and then annealed at 350 °C for 2 h to obtain pristine-$TiO_2$/TM.

**Preparation of Vo-TiO₂.** The pristine-$TiO_2$ array was spread on the quartz boat in a plasma reactor. Afterwards, the pristine-$TiO_2$ array was treated using a homemade plasma generator with a tube furnace and a commercial 13.56 MHz radio frequency source under 99.99% $H_2$ flow (200 sccm). For the plasma treatment, the radio frequency power was 300 W for 5 min, and the desired Vo-$TiO_2$ array was obtained. The loading of Vo-$TiO_2$ was evaluated by a high-precision analytical balance instruction and was determined to be 1.6 mg cm$^{-2}$.

**Electrocatalytic HER measurements.** The electrochemical HER measurements were performed in a typical three-electrode system by using a CHI 604E electrochemical workstation, in which a Ag/AgCl electrode acts as the reference electrode, pristine $TiO_2$/TM or Vo-$TiO_2$/TM as the working electrode, and a platinum sheet as the counter electrode, respectively. All experiments were conducted at room temperature (≈25 °C). The exposed active area for electrocatalysis was 1.0 cm × 1.0 cm, and LSV was recorded at a scan rate of 5 mV s$^{-1}$ in 0.1 M $H_2SO_4$ electrolyte (pH = 1.0).

**In situ EPR spectroelectrochemical measurements.** In situ EPR spectroelectrochemical measurements were carried out with BRUKER A 300 operating at X-band frequency of 9.6 GHz under ambient temperature[95]. A three-electrode spectroelectrochemical cell was employed, with an externally applied potential to drive electrochemical processes. The $TiO_2$ or $V_O$-$TiO_2$ modified ITO electrodes by the drop coating method were used as working electrodes, while Ag/AgCl and Pt wires were used as the reference electrode and counter electrode, respectively. The relevant experimental setup is illustrated in Supplementary Fig. 35. The microwave power supplied to the resonator was maintained at 4 mW to avoid EPR signal saturation. A magnetic field modulation frequency of 100 kHz was applied with a modulation amplitude of 0.1 mT or lower to prevent spectral distortion from over-modulation. Data acquisition parameters were set with a conversion time of 40.96 ms and a time constant of 20.48 ms.

**Electrochemical in situ NAPXPS measurements.** In situ NAPXPS measurements were performed at the BL02B01 beamline of the Shanghai Synchrotron Radiation Facility (SSRF)[96]. A custom-designed in situ electrochemical cell, similar to those reported in previous studies[97,98], was employed to investigate the electrode–electrolyte interface under controlled potential conditions. The cell was equipped with a Nafion proton-exchange membrane to separate the vacuum environment from the liquid electrolyte (0.1 M $H_2SO_4$), while maintaining proton conductivity. Before electrochemical measurements, the Nafion 117 membrane was first protonated in 5 wt% $H_2O_2$ for 1 h, then treated in 0.5 M $H_2SO_4$ for 3 h, and finally immersed in water for 6 h. All steps were performed at 80 °C. A continuous and uniform working electrode was fabricated by spraying $TiO_2$ ink onto the surface of the activated Nafion 117 membrane. The ink was prepared by sonicating a mixture of graphite powder, $TiO_2$ nanoparticles, Nafion ionomer, and absolute ethyl alcohol for 30 min. A clean Pt foil was used as the counter electrode and a home-made Ag/AgCl wire as the reference electrode. The sample was grounded to the analyzer to ensure a well-defined potential reference. The electrochemical potential was controlled using a potentiostat (BioLogic SP-200), and Ti 2p spectra were collected at the photon energy of 1240 eV as a function of applied potential, starting from the open-circuit voltage (OCV) to −2.2 V$_{SHE}$. All spectra were energy-referenced using the C 1s from the graphite powder for checking charging effects. The pressure in the analysis chamber was maintained at 0.2 mbar (water vapor), which also served as the reactant and minimized electrolyte evaporation. XPS spectra were fitted using the GL(30) line shape and Shirley-type background by CasaXPS software.

**Electrochemical in situ Raman spectroscopy measurements.** In situ Raman measurements were carried out on an XploRA PLUS Raman spectrometer equipped with a 50×objective and a 638 nm He-Ne laser. The laser filter was set at 20%, while the retention time was set to 20 s to obtain each spectrum. The Raman spectrometer was calibrated using a Si wafer. All spectra were recorded using a commercial three-electrode spectro-electrochemical cell. The as-prepared materials, Ag/AgCl (3.0 M KCl), and Pt slice were employed as the working, reference, and counter electrode, respectively. Relevant measurements were conducted in 0.1 M $H_2SO_4$ electrolyte.

**In situ EFM characterizations.** In situ EFM experiments were performed on a commercial atomic force microscope (AFM, Dimension Icon, Bruker) using a conductive AFM probe (SCM-PIT-V2, Pt/Ir-coated, Bruker). Before the EFM measurements, to induce local charges at different points on the $TiO_2$ film, the AFM probe was biased with different voltages and brought into contact with the sample at discrete points. The EFM was operated under a dual-pass tapping mode. On the first pass, the topographical image was obtained without applying any voltage between the probe and the sample. On the second pass, an external bias was applied to the conductive probe as it was lifted by a constant height of 20 nm above the sample surface. The phase shift

signal of the cantilever was fed into two lock-in amplifiers to demodulate the 1ω and 2ω components.

## Data availability

Full data supporting the findings of this study are available within the article and its Supplementary Information as well as from the corresponding authors upon request. Source data are provided with this paper. The computationally optimized adsorption and transition state structures are freely available in the Zenodo data repository at https://doi.org/10.5281/zenodo.18210690. Source data are provided with this paper.

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

## Acknowledgements

T.W. was supported by the National Natural Science Foundation of China (Nos. 52202214, 52573243, and 11227902) and China National Post-doctoral Program for Innovative Talents (No. BX2021053). L.L. was supported by the National Natural Science Foundation of China (Nos. 52225308 and 52533010). M.M.M. and K.H. gratefully acknowledge support by the Academy of Finland (grant numbers 317739 and 338228), and the Jane and Aatos Erkko Foundation (funding to the LACOR project). The numerical calculations in this paper have been done on the Computing Center in Xi'an. Supported by the Center for HPC, University of Electronic Science and Technology of China. The authors thank the BL02B01 beamline of the Shanghai Synchrotron Radiation Facility (SSRF) for providing the XPS beamtime.

## Author contributions

T.W. designed and performed simulations and experiments. T.W. and X.G performed in situ EPR and Raman spectroelectrochemistry measurements. G. Z. performed in situ EFM measurements. T.W., X.G and L. S. performed the electrochemical tests. S.S and H.Z. performed in situ NAPXPS tests. M.M.M. and K. H. supervised the simulations. T.W., Y.Z., Z.L., L.L., M.M.M., and K. H. contributed to analyzing the results and writing the manuscript.

## Competing interests

The authors declare no competing interests.
