## [Transparent Peer Review file · Nature Communications]

Potential-dependent polaron formation activates TiO₂ for the hydrogen evolution reaction

Corresponding Author: Professor Karoliina Honkala

Version 0:

Reviewer comments:

Reviewer #1

(Remarks to the Author)

Potential-dependent polaron formation activates TiO₂ for the hydrogen evolution reaction

The work by Honkala and coworkers on the in-situ formation of polarons, stimulated by an external electrode potential, is an interesting read and definitely of relevance to the broad readership of Nature Communications. The implications of polarons for semiconductor electrocatalysis is novel, and I strongly recommend this work for publication.

However, before publication can take place, I ask the authors to address the following criticism:

i) Introduction: the authors introduce the conventional approach of using descriptor-based analysis, which connects a material's property to the electrocatalytic activity. While the chosen references (e. g., 30-31 for scaling relations) appear to be generic (I suppose that reference 31 refers to thermal catalysis rather than electrocatalysis), aren't there particular works that have focused on the scaling relations of electrocatalytic processes such as the HER on semiconductors such as TiO₂ (<https://doi.org/10.1002/cctc.201100160>)? Here, I believe that the authors could strengthen the discussion and motivation of their work through proper references.

On the other hand, are the works that have reported (potential-dependent) scaling relations by analyzing the adsorption energetics in dependence of the applied electrode potential? I believe that also in this context, the authors should carefully scan the literature, because this might enable them to strengthen the discussion and motivation of their work.

ii) I am unsure whether the example of Co₃O₄ is well chosen for the introduction of polarons because it is known that Co₃O₄ transforms into CoOOH for sufficiently large anodic potentials (<https://doi.org/10.1038/ncomms9625>). Therefore, the formation of Co⁴⁺ could also be related to a different process under the conditions of U > 1.5 V vs. RHE.

iii) How do the authors determine the potential of zero charge (PZC) of U = 1.0 V_{SHE}?

iv) "We also note that the Tafel slope of VO-TiO₂ at lower overpotentials is close to the cardinal value of 60 mV/dec, commonly considered as an indicator of HER following the Volmer-Heyrovsky mechanism, with the Heyrovsky step identified as the rate-determining step.⁷⁵"

I disagree with this statement, and I also believe that the paper by Shinagawa (ref. 75) tells a different story.

The cardinal value of about 60 mV/dec. indicates that the apparent transfer coefficient is equal to 1 (β = 1). Thus, there is one electron transferred before (quasi-equilibrium) the transition state with highest free energy (γ = 1), and the transition state with highest free energy is related to a chemical step (α = 0) that shows no dependency on the applied electrode potential.

The above discussion refers to the classical analysis based on the Butler-Volmer formalism. To my opinion, the only option to relate a Tafel slope of 60 mV/dec. to a HER mechanism is by adopting the Volmer-Tafel mechanism because this is the only HER mechanism comprising a chemical (Tafel) step:

*H + * + H⁺ + e⁻ → *H + *H (pre-equilibrated, γ = 1)

*H + *H → * + H₂ (transition state with highest free energy ⇒ RDS; α = 0)

β = γ + α = 1 Tafel slope ⇒ 60 mV/dec.

In contrast, for the Volmer-Heyrovsky mechanism, Tafel slopes of 120 mV/dec. and 40 mV/dec. are expected using the classical framework.

Here, it should be noted that the classical Butler-Volmer analysis does not account for

- Coverage effects that can impact the rate of electrocatalytic reactions (this is essentially the message of the work by Shinagawa et al.; quasi-equilibrium in the BV formalism tacitly assumes that the coverage is constant, and steady-state approaches are called for to enable modulation of the coverage as a function of U, which can impact the Tafel slope; for a further discussion, the authors are related to the following work: <https://doi.org/10.1002/elsa.202100037C>)

- The discussion of chemical reaction steps is based on a canonical approach because it is tacitly assumed (in the classical approach; see J. O'M. Bockris, A. K. N. Reddy, *Modern Electrochemistry*, Vol. 2, Plenum Publishing Corporation, New York, 1973, pp 991) that the transition-state free energy of a chemical step does not depend on the applied electrode potential. Both notions are not necessarily fulfilled, and I believe that the authors could expand the discussion in this regard and use these (probably unfulfilled) connections in their work.

v) "the Volmer reaction ($\text{H}_3\text{O}^+ + \text{e}^- \rightarrow \text{H}^* + \text{H}_2\text{O}$), 76-78 is very facile and cannot be the rate-determining step"

Please provide a definition of the 'rate-determining step' (RDS) in the manuscript. Using the classical approach (or even using a steady-state approach), the RDS is not necessarily ascribed to the elementary step with the highest barrier, dG_a , as plotted in Figure 5. Therefore, I would advise to plot free-energy diagrams that comprise the entire information on the thermodynamics and kinetics of the HER over TiO_2 .

Based on the above discussion (cf. iv), it appears to be relevant that the authors also investigate the kinetics of the Tafel step. I believe that this enables exacerbating the discussion in the manuscript because the chosen grand canonical framework allows comprehending whether the barrier of the Tafel step is potential dependent. Hence, the authors can scrutinize another fundamental assumption of the classical picture for semiconductor electrocatalysis.

vi) Figure 5: there is a typo in the figure (Volmer instead of "Vomler"). Figure 5d is not a Butler-Volmer-like relationship per se, but this plot allows determining the transfer coefficient (α) for the electrochemical Volmer step. It would be interesting to discuss the transfer coefficient and compare it to the conventional assumption of $\alpha = 0.5$ for an electrochemical reaction step. What are the transfer coefficients for the Heyrovsky or Tafel steps?

vii) If the BEP relationship in Figure 5e holds, would it be possible to use only the thermodynamic picture to derive conclusions on the activity of (V_{O})- TiO_2 in the HER? Here, free-energy diagrams are helpful to solve this scientific question of interest to the community, considering that most computational works rely on the thermodynamic picture of adsorption free energies.

viii) The transition-state free energies of the Volmer and Heyrovsky steps in the HER are reduced to a different extent with increasing cathodic overpotential. The authors specified values of 1.01 eV and 1.10 eV in Figure 5c, respectively. This might indicate that at sufficiently large cathodic overpotentials, the Heyrovsky step will no longer limit the rate, but instead the Volmer step governs the electrocatalytic activity.

I believe that this point should be discussed in more detail, which is possible by constructing free-energy diagrams along the reaction coordinate.

ix) Ultimately, the construction of free-energy diagrams allows translating the calculated energetics to a Tafel plot, which can be directly compared to Figure 4. In case of the Volmer-Heyrovsky mechanism, the corresponding equations can be found in <https://doi.org/10.1002/elsa.202100037C>.

x) "By increasing the potential to the experimental onset potential of -0.3 VSHE,"

What is the 'so-called' onset potential? According to Compton, this term is ill-defined

(<https://doi.org/10.1016/j.apmt.2019.05.011>), unless the chosen potential is clearly related to a current density, which is not the case in the discussion here.

xi) It would be desirable if the grand canonical barriers were marked with a "#", as this allows an easy distinction between the thermodynamic and kinetic quantities.

xii) Pages 17/18: the discussion of the RDS should be refined based on the comments above.

xiii) Page 18 "Impact of potential-dependent polaron formation on free energy relations, kinetics, and electrocatalyst design": the discussion of the traditional Butler-Volmer formalism should be refined based on additional calculations and the comments above.

xiv) The surface coverage of TiO_2 is dependent on the applied electrode potential. How do the authors ensure that they have chosen the 'correct' coverage under HER conditions for the modeling of the HER thermodynamics and kinetics?

I did not see that the authors determined surface coverages as a function of U. However, the omission of surface coverages can lead to erroneous results on the electrocatalytic activity (<https://doi.org/10.1039/D4CP01792G>). Therefore, I ask the authors to address this point seriously.

Reviewer #2

(Remarks to the Author)

In this manuscript, Honkala and co-workers reported potential-dependent polaron formation and its impact on electrocatalytic HER with TiO_2 semiconductor electrodes. The detailed theoretical calculations and experimental results well support their

view. The results are interesting and the data presented are ample. Therefore, this work can be considered for publication in Nature Communications after some revisions.

1. The author's description of the order of the pictures is very confusing, making it difficult to read the corresponding content smoothly. Please reorganize the order of the pictures and texts for reading easily. All the data in Supporting Information should be mentioned and labeled in the Manuscript.
2. The author mentioned the corresponding charge value changes when describing the DOS in Figure 1. Based on the reviewer's understanding, only the DOS can qualitatively determine the electron distribution position. Please explain how to determine the specific value of the change of corresponding charges, and suggest that it be clearly marked in figures.
3. The reviewer noticed that the author mentioned that the AMM of $\text{Ti}_5\text{c}3+$ is stable at $U=-2$ V, but there were still several shaking points, and as increased to $U=-3$ V, the shaking became more intense. How to explain the cause of the shaking points? Or is there a charge transfer in the kinetic process?
4. The CIP-DFT method is a very important research method for analyzing electrochemical systems. The reviewer suggested that the key points of the CIP-DFT method be briefly described in the calculation details of the main text, which is conducive to its promotion and application.
5. The O 1s and Ti 2p XPS spectra should be provided.
6. CV tests need to be performed.
7. In Table S9, there is no blue or green text.

Reviewer #3

(Remarks to the Author)

In this article, the authors studied the formation and change of the Ti^{3+} and oxygen vacancy on TiO_2 in the (photo)electrocatalytic process using in situ experiments. Despite the wealth of experimental and theoretical calculations, the novelty of the conclusions and introduction regarding Ti^{3+} and oxygen vacancies are insufficient, and a large experimental literature already exists. We therefore recommend that the authors transfer to a more specialized journal.

Reviewer #4

(Remarks to the Author)

The authors, from a strong collaboration among different theoretical groups, report a thorough study on potential-dependent changes of semiconducting electrodes. While electrode/catalyst surface restructuring under working conditions is actively studied experimentally, their atomic structure remains unclear. This paper, focused on density functional theory-based calculations, provides noteworthy results of significance to the field. However, the involved experimental work and its link with the theoretical results need further development to fully support the conclusions. To substantiate the novelty of this work as stated in the abstract "combining calculations with multiple in situ experiments", major revisions are required before publication, especially for the broad audience of the journal.

In the in-situ UV-Vis spectra (Fig. 2d), the absorbance peak around 630 nm is poorly defined at -2.0 V_{SHE} but more pronounced at -2.2 V_{SHE}. Why the calculation (Fig. 2a) was not carried out at -2.2 V_{SHE} should be justified. Since 630 nm corresponds to 1.97 eV, should the label in Fig. 2d " Ti^{3+} polaron" be changed to " Ti_{5C^3+} " to link to Fig. 2a? In the in-situ EPR spectra (Fig. 2e), no experimental data is provided at potentials more negative than -2.4 V_{SHE}. How does this relate to the calculation (Fig. 2c) at -3.0 V_{SHE}? In the in-situ EFM maps (Fig. 2f-h), potentials are measured in solid-state (dry) conditions. How do they relate to the electrochemical (wet) conditions used in the calculations (Fig. 2a-c)? Referring to Fig. 1, it is understandable that -2.0 and -3.0 V_{SHE} were chosen from modelling, but experimental support is insufficient in Fig. 2 especially when potential is key to this work.

The disconnection between experiments and theory is also evident in the SI. In the in-situ EXAFS spectra (Fig. S9), how does the Ti-O bond change relate to the calculations (Fig. 2a-c)? As the main claim of this paper is that the applied potential alters the electronic state of the electrode surface, in-situ Raman spectroscopies of TiO_2 should be potential-dependent (see examples in Energy Environ. Mater. 2024, 7, e12692 and ACS Appl. Nano Mater. 2023, 6, 6528). The unchanged spectra (Fig. S37) contradict this claim. They should be enlarged to show intensity and position changes with potential.

In the introduction, an overview of the latest theoretical methods and results from the established literature is missing. This makes it difficult to compare the theoretical work with the state-of-the-art. The current overview of the experimental methods should be compressed to enhance relevance to the calculations. The introduction to the polaron and defects, as the focus of this work, is scattered across pages 4 and 5 and should be rearranged and brought forward.

In the methods, more experimental details should be provided to ensure reproducibility. For example, the composition of the Ag/AgCl electrode, as the reference electrode, changes depending on whether 1M, 3M, or saturated KCl is used. The manufacture and model of electrodes and cells should be specified; if home-made, photos or schematics with parameters should be provided in the SI. This is crucial as in-situ measurements are often restricted by limited space (e.g., EPR). Furthermore, the laser power for Raman spectroscopy should be provided.

Throughout the manuscript, abbreviations should be defined when they first appear, e.g., $\text{Vo-TiO}_2(101)$.

Reviewer comments:

Reviewer #1

(Remarks to the Author)

The authors have made an extensive revision and I am convinced about the importance of this work to the field. I would definitely stress the result of an apparent transfer coefficient larger than 1 because this is highly novel and may have implications beyond the discussed example of TiO₂ for HER. I fully support this work for publication in Nature Communications.

Reviewer #2

(Remarks to the Author)

The authors have addressed my questions and concerns. And I think I am fine with the acceptance for the manuscript.

Reviewer #3

(Remarks to the Author)

The reviewer has read the revised manuscript and confirmed that careful revisions have been made. I can now recommend the revised manuscript for publication.

Reviewer #5

(Remarks to the Author)

In the revised manuscript, the authors have introduced in-situ EPR spectroelectrochemistry, in-situ XAS, and in-situ Raman analyses to support their claim regarding the formation of surface Ti^{5c3+} polarons during the reaction. However, these additions raise several critical concerns that might significantly impact the conclusions and overall claims of this manuscript. The following issues must be addressed before the manuscript can be considered for publication:

1. The in-situ EPR spectroelectrochemistry conducted on TiO₂-modified ITO electrode in 0.1 M H₂SO₄ electrolyte demonstrates the ability to detect surface Ti^{5c3+} species at room temperature, suggesting their lifetime ranges from milliseconds to seconds. In contrast, the CIP-DFT-MD simulations indicate a much shorter timescale in the picosecond range. This significant mismatch must be reconciled, and the applicability of the experimental and theoretical approaches to each other should be clearly explained.
2. If the presence of surface Ti^{5c3+} polarons during reaction is to be substantiated, surface-sensitive in-situ techniques—such as soft XAS (Ti L-edge) or in-situ XPS—should be utilized. The current reliance on Ti K-edge XAS, a bulk-sensitive method, is insufficient to confirm the formation of surface-localized species.
3. The Ti K-edge XAS data show negligible shifts in the absorption edge and EXAFS features, which do not convincingly support the presence of surface Ti^{5c3+} polaron. This stands in contrast to the significant EPR signal, which implies a considerable population of such surface states, which may be detected. The inconsistency between the two techniques must be resolved.
4. A comparative analysis of the Ti K-edge XAS spectra for pristine TiO₂ and VO-TiO₂ should be provided to clarify any differences in Ti oxidation states and local environments. This comparison is essential for substantiating claims regarding Ti state modulation.
5. The in-situ XAS measurements were conducted at -1.0 and -2.0 VSHE, yet in-situ EPR data suggest no significant Ti state variation until -2.4 VSHE. To enable meaningful correlation between the two datasets, XAS measurements should be performed at potentials consistent with those used in EPR analysis.
6. The EXAFS fitting lacks sufficient resolution to draw meaningful conclusions. The integral deviation in coordination numbers implies high uncertainty, and bond distance values with only two significant figures do not indicate any appreciable structural change during HER. The authors should provide a more robust fitting analysis with improved statistical confidence.
7. The in-situ Raman analysis focuses narrowly on a single E_g peak, which predominantly reflects bulk TiO₂ properties. A thorough examination of all characteristic Raman modes (A_{1g}, B_{1g}, and E_g) is necessary to assess potential structural or electronic changes during HER. Additionally, the reported fluctuation of the E_g peak—attributed to varying oxygen vacancy concentrations—is not corroborated by XAS data, further weakening this interpretation.

In summary, while the newly added in-situ characterization techniques enrich the study, the data presented raise important questions that must be rigorously addressed. Clarifying these points will be crucial to determining the scientific validity and publication readiness of the manuscript.

Version 2:

Reviewer comments:

Reviewer #5

(Remarks to the Author)

In this revised manuscript, the authors have addressed several of the initial concerns; however, a critical issue raised: whether the spectroscopic changes observed during in-situ measurements are attributed to the formation of intermediates (e.g., polarons) under catalytic conditions or to irreversible alternations in the titanium oxidation state induced by the application of high negative potentials. This distinction is central to the authors' main claim—that polarons are generated in situ in a reversible manner. Given this, further clarification and supporting evidence are essential before the manuscript can be considered for publication.

1. The reviewer appreciates the authors' explanation in the response letter regarding the time scale discrepancy between the experimental EPR measurements and theoretical CIP-DFT-MD simulations. However, the manuscript itself contains only a brief mention of this limitation. Since such divergence could mislead readers unfamiliar with the constraints of time-resolved simulations, a more detailed explanation—akin to what was provided in the response—should be incorporated into the main text.

2. The current in-situ Ti K-edge XAS measurements, which primarily probe bulk states, are insufficient to conclusively establish surface-localized Ti^{3+} formation during electrocatalysis. The authors did not conduct in-situ surface-sensitive techniques (e.g., Ti L-edge XAS or near-ambient pressure XPS), instead offering ex-situ Ti L-edge XAS results without detailed experimental description.

This omission raises concerns: if any change is observed in ex-situ measurements, it could represent irreversible structural modification rather than the presence of reaction intermediates. Such changes might result from the application of high negative bias rather than the catalytic process itself, contradicting the manuscript's central claim of reversible in-situ polaron formation.

To address this, the authors must:

- Provide control measurements taken under open-circuit conditions after high-voltage application across all characterization techniques (EPR, XAS, Raman, and EFM) to determine whether observed changes are reversible.
- Clarify whether the Ti L-edge XAS was indeed ex-situ and, if so, provide full methodological details.
- Correct the assertion regarding the presence of a pre-edge feature in Ti L-edge XAS, which is not physically supported due to the nature of the core-hole-d-electron interactions. This suggests a fundamental misunderstanding of the technique's operational principles.

3. To align with the high standards expected by Nature Communications, the reviewer maintains that it is essential to perform in-situ Ti L-edge XAS or in-situ near-ambient pressure XPS to directly probe surface electronic and chemical states under operational conditions. These techniques are currently available and would provide critical evidence to support the manuscript's central claim regarding the in-situ and reversible formation of polarons.

In addition, the authors are required to include control measurements under open-circuit conditions after voltage application. This is crucial to distinguish between reversible changes associated with catalytic intermediates and irreversible modifications induced by the application of high bias. Without such controls, it remains unclear whether the observed spectroscopic changes truly reflect catalytic dynamics or are artifacts of material degradation.

4. The authors highlight a 0.23 eV shift in the Ti K-edge XANES region to infer valence state changes. However, considering the large energy difference between Ti^0 (~4965 eV) and Ti^{4+} (~4983 eV), a shift of this magnitude is negligible and corresponds to a minor oxidation state change (~0.05 e^-), which may not yield meaningful chemical insight. The authors are requested to perform quantitative linear fitting based on the edge inflection point to determine Ti oxidation states more accurately and re-evaluate whether such a minor energy shift can meaningfully support the claims of in-situ polaron formation.

The validity of the EXAFS fitting also warrants further scrutiny:

- The fitting models used for $Vo-TiO_2$ and pristine TiO_2 must be explicitly described, including structural parameters and assumptions.
- Experimental spectra and fitted curves should be presented together to demonstrate fit quality.
- High R-factors (approaching 0.02) diminish the reliability of subtle changes in coordination number and bond distances. More robust evidence is required to validate such interpretations.
- Notably, ΔE_0 values appear identical across TiO_2 and $Vo-TiO_2$ samples, which is inconsistent. ΔE_0 is a variable that reflects differences in electronic environment and should be independently optimized for each sample.

5. From the authors' response and images provided, there are concerns about the suitability of the in-situ XAS setup:

- The explanation that bubble formation interferes with XAS measurements is questionable, as the in-situ cell is designed for back-side illumination, where the X-ray beam reaches the catalyst layer without passing through the electrolyte, minimizing bubble interference.
- The positioning of the catalyst relative to the Lytle detector appears incorrect. In standard setups, the sample should be positioned near the center of the fan-shaped detection grid (approximately one-third from the edge), regardless of left/right alignment. Misalignment here may compromise data integrity.

These issues cast doubt on the reliability of the in-situ EXAFS results presented.

To uphold the rigorous standards of Nature Communications, the authors must provide compelling evidence that the observed spectral changes reflect reversible, in-situ generation of polarons rather than irreversible structural or compositional alterations induced by high-voltage application. Addressing the methodological limitations and incorporating appropriate control and surface-sensitive measurements are essential steps toward strengthening the manuscript's conclusions.

Version 3:

Reviewer comments:

Reviewer #5

(Remarks to the Author)

The authors have addressed the reviewer's suggestions and concerns well. The reviewer recommends its publication in the current version.

Reviewer #1

Potential-dependent polaron formation activates TiO₂ for the hydrogen evolution reaction

The work by Honkala and coworkers on the in-situ formation of polarons, stimulated by an external electrode potential, is an interesting read and definitely of relevance to the broad readership of Nature Communications. The implications of polarons for semiconductor electrocatalysis is novel, and I strongly recommend this work for publication.

However, before publication can take place, I ask the authors to address the following criticism:

We appreciate your positive feedback on our work. We also sincerely thank the referee for the exceptionally detailed and helpful comments, which have allowed us to improve our work.

(1) Introduction: the authors introduce the conventional approach of using descriptor-based analysis, which connects a material's property to the electrocatalytic activity. While the chosen references (e. g., 30-31 for scaling relations) (I suppose that reference 31 refers to thermal catalysis rather than electrocatalysis), aren't there particular works that have focused on the scaling relations of electrocatalytic processes such as the HER on semiconductors such as TiO₂ (<https://doi.org/10.1002/cctc.201100160>)? Here, I believe that the authors could strengthen the discussion and motivation of their work through proper references. On the other hand, are there works that have reported (potential-dependent) scaling relations by analyzing the adsorption energetics in dependence of the applied electrode potential? I believe that also in this context, the authors should carefully scan the literature, because this might enable them to strengthen the discussion and motivation of their work.

Based on your suggestions, we have updated the references to more relevant ones to improve the discussion and motivation of our work, please see page 3 for reference 5. Unfortunately, we could not find any scaling relation for HER on oxides, computed using either GCE-DFT or the computational hydrogen electrode method. However, the BEAST database has a large collection of GCE-DFT -computed hydrogen adsorption energies on metallic electrodes. The results show that depending on the electrode material, the hydrogen adsorption energies depends either linearly or more generally quadratically on the applied potential. While the linear relation is similar to CHE, the quadratic dependency requires the use of GCE-DFT.

We have carefully examined the scaling relationship between hydrogen adsorption energy and electrode potential for the studied TiO₂ electrodes. Two distinct potential-dependent regions are identified. In region 1 (0 to -0.5 V vs SHE), the hydrogen adsorption energy decreases as the potential becomes more negative. In region 2 (negative of -0.5 V vs SHE), the adsorption energy *increases* with more reducing potentials, see Figure R1a. Region 1 has a constant polaron concentration (schematically in Figure R1b), whereas in region 2 the polaron concentration increases (schematically in Figure R1c). These observations suggest that the potential-dependent polaron formation breaks the linear or quadratic relationship between hydrogen adsorption energy

and electrode potential observed for metallic electrodes [reference 33 in main text]. Therefore, for semiconductor electrodes, the potential-dependent relationship need to be carefully considered and possibly revised, especially in cases involving polaron formation.

Figure R1. (a) Scaling relationships between hydrogen adsorption energy and the electrode potential. (b,c) The top view of the defected surface. The dashed black circle represents a missing O_{br} atom. Ti cations are grey, O anions are red, and blue stands for the Ti^{3+} cation with a polaron state.

Figure R1 and above discussion has been added into the main manuscript on pages 17-18 in blue.

(2) I am unsure whether the example of Co_3O_4 is well chosen for the introduction of polarons because it is known that Co_3O_4 transforms into $CoOOH$ for sufficiently large anodic potentials (<https://doi.org/10.1038/ncomms9625>). Therefore, the formation of Co^{4+} could also be related to a different process under the conditions of $U > 1.5$ V vs. RHE.

We agree that Co_3O_4 is a somewhat problematic example as the reasons for its activity are not fully understood. Also, our simulations of the $Co_3O_4(110)$ at different magnetic states does not conclusively demonstrate polaron formation (Figure R2).

Therefore the example of Co_3O_4 has been removed from the introduction of the main manuscript and replaced with a more relevant example of Pt_1/CeO_2 , *please see the text highlighted in blue on page 4, in the main manuscript*.

Figure R2. (a-c) The changes in the electronic state of surface Co with electrode potential. (d-f) The projected density of states of Co atoms at different spin states.

(3) How do the authors determine the potential of zero charge (PZC) of $U = 1.0 \text{ V}_{\text{SHE}}$?

The potential of zero charge (PZC) was identified by studying how the number of excess electrons (surface charge) in the TiO_2 system varies with the electrode potential. The results indicate that the PZC is located at $U = 1.0 \text{ V}_{\text{SHE}}$, see Figure R3.

The determination of potential of zero charge is now explained in SI on page 12 and Figure R3 is given as Figure S7 in the SI.

Figure R3. The change of excess electrons as function of electrode potentials.

(4) “We also note that the Tafel slope of VO-TiO₂ at lower overpotentials is close to the cardinal value of 60 mV/dec, commonly considered as an indicator of HER following the Volmer-Heyrovsky mechanism, with the Heyrovsky step identified as the rate-determining step.⁷⁵” I disagree with this statement, and I also believe that the paper by Shinagawa (ref. 75) tells a different story.

The cardinal value of about 60 mV/dec. indicates that the apparent transfer coefficient is equal to 1 ($\beta = 1$). Thus, there is one electron transferred before (quasi-equilibrium) the transition state with highest free energy ($\gamma = 1$), and the transition state with highest free energy is related to a chemical step ($\alpha = 0$) that shows no dependency on the applied electrode potential.

The above discussion refers to the classical analysis based on the Butler-Volmer formalism. To my opinion, the only option to relate a Tafel slope of 60 mV/dec. to a HER mechanism is by adopting the Volmer-Tafel mechanism because this is the only HER mechanism comprising a chemical (Tafel) step:

$$\beta = \gamma + \alpha = 1 \Rightarrow \text{Tafel slope } \Rightarrow 60 \text{ mV/dec.}$$

In contrast, for the Volmer-Heyrovsky mechanism, Tafel slopes of 120 mV/dec. and 40

mV/dec. are expected using the classical framework.

Here, it should be noted that the classical Butler-Volmer analysis does not account for

- Coverage effects that can impact the rate of electrocatalytic reactions (this is essentially the message of the work by Shinagawa et al.; quasi-equilibrium in the BV formalism tacitly assumes that the coverage is constant, and steady-state approaches are called for to enable modulation of the coverage as a function of U, which can impact the Tafel slope; for a further discussion, the authors are related to the following work: <https://doi.org/10.1002/elsa.202100037C>)

- The discussion of chemical reaction steps is based on a canonical approach because it is tacitly assumed (in the classical approach; see J. O'M. Bockris, A. K. N. Reddy, Modern Electrochemistry, Vol. 2, Plenum Publishing Corporation, New York, 1973, pp 991) that the transition-state free energy of a chemical step does not depend on the applied electrode potential.

Both notions are not necessarily fulfilled, and I believe that the authors could expand the discussion in this regard and use these (probably unfulfilled) connections in their work.

We agree that our initial analysis of the Tafel slope of 60 mV/dec was flawed. The analysis presented by the reviewer offers an excellent framework for interpreting the Tafel slope and after careful consideration, we have adopted this approach in our revised analysis and **have included it in the Supporting information Section S10.**

Additionally, we performed new calculations that clearly demonstrate that the Tafel step is not only the RDS, but also exhibits strong potential dependency, as detailed below. On the V₂₀-TiO₂ surface, the surface oxygen atoms readily interact with H atoms to form hydroxyl group and we examined the adsorption of a hydrogen atom on the oxygen atom nearest to the oxygen vacancy, which leads to OH group without a barrier through a Volmer step ① in Figure R4a (Figure S17 in the revised SI). Our CIP-DFT structure relaxation and NEB results indicate that the second Volmer step involves the nearest oxygen atom adjacent to the vacancy, forming an OH functional group. This process leads to the spontaneous, barrierless dissociation of H₃O⁺ to yield a geometry where a hydrogen atom directly adsorbs at the V_O site located between two neighboring Ti³⁺ atoms. This step is depicted as step ② in Figure R4a.

Subsequently, the two adsorbed hydrogen atoms combine to form molecular hydrogen via the “chemical” Tafel step (see step 3 in Figure R4a), which exhibits a relatively high and strongly potential-dependent barrier (see Figure R4b); we identify the Tafel step as **the rate-determining step (RDS) on V₂₀-TiO₂.** Overall, our simulations support the Volmer-Tafel mechanism as the operative HER pathway, where *H + * + H⁺ + e⁻ → *H + *H step is fast and pre-equilibrated, while *H + *H → * + H₂ is the rate-determining “chemical” step

Figure R4. (a) A HER process on the V₂O-TiO₂ surface. (b) The activation energy of a Tafel step as a function of a potential on the V₂O-TiO₂ surface.

We have revised the text in the main manuscript starting at the bottom of page 15 and continuing to page 16, where we write as follows: *“Therefore, we considered the impact of a double oxygen vacancy Thus Tafel step appears to be the rate-determining step on the V₂O-TiO₂ (101) surface as it has the highest activation energy.”* Discussion of further details can be found from Figure S17 in Supporting Information page 23.

(5) “the Volmer reaction ($\text{H}_3\text{O}^+ + \text{e}^- \rightarrow \text{H}^* + \text{H}_2\text{O}$), 76-78 is very facile and cannot be the rate-determining step” Please provide a definition of the ‘rate-determining step’ (RDS) in the manuscript. Using the classical approach (or even using a steady-state approach), the RDS is not necessarily ascribed to the elementary step with the highest barrier, dG_a , as plotted in Figure 5. Therefore, I would advise to plot free-energy diagrams that comprise the entire information on the thermodynamics and kinetics of the HER over TiO₂.

Based on the above discussion (cf. iv), it appears to be relevant that the authors also investigate the kinetics of the Tafel step. I believe that this enables exacerbating the discussion in the manuscript because the chosen grand canonical framework allows comprehending whether the barrier of the Tafel step is potential dependent. Hence, the authors can scrutinize another fundamental assumption of the classical picture for semiconductor electrocatalysis.

Please refer to the response provided in question (4).

(6) Figure 5: there is a typo in the figure (Volmer instead of “Vomler”). Figure 5d is not a Butler-Volmer-like relationship per se, but this plot allows determining the transfer coefficient (alpha) for the electrochemical Volmer step. It would be interesting to discuss the transfer

coefficient and compare it to the conventional assumption of $\alpha = 0.5$ for an electrochemical reaction step. What are the transfer coefficients for the Heyrovsky or Tafel steps?

We calculated the transfer coefficients of the Volmer steps and the Tafel step as presented in the new section S1.3 in supporting information. According to the formula $\Delta G^\ddagger = \Delta G_0 - \alpha eU$, where α is the transfer coefficient, we found that for the Volmer step, $\alpha = 1.42$ and 1.45 for the first and second Volmer step, respectively. (Figure R5a). For the Tafel step, $\alpha = 1.22$ (Figure R5b). These results suggest that the electrode potential plays a crucial role in lowering the barrier for both the electrochemical Volmer step and “chemical” Tafel step.

The underlying reason for the transfer coefficient α to be larger than one is the concomitant and potential-dependent polaron formation and electron transfer in HER steps. This suggests that future research on the mechanism of semiconductor electrocatalysis would benefit from incorporating more systematic analysis.

Figure R5. (a) The grand canonical barriers for Volmer reaction as a function of electrode potential on V₂₀-TiO₂ and Au. (b) The grand canonical barriers for Tafel step as a function of electrode potential on V₂₀-TiO₂.

The transfer coefficient values are now give on page 29 in SI and Figure R5 is presented as Figure S22 in SI.

(7) If the BEP relationship in Figure 5e holds, would it be possible to use only the thermodynamic picture to derive conclusions on the activity of (V₂O)-TiO₂ in the HER? Here, free-energy diagrams are helpful to solve this scientific question of interest to the community, considering that most computational works rely on the thermodynamic picture of adsorption free energies.

Our calculations indicate that both the Tafel step in the Volmer-Tafel mechanism and the Volmer step in the Volmer-Heyrovsky mechanism on the V₂₀-TiO₂ surface follow the BEP relationship, see Figure R6. Consequently, it may be possible to assess HER performance using solely the thermodynamic perspective. However, due to the current limitations of the CIP-DFT

method in accurately modeling potential-dependent processes across the band gap, the activation energy for the Heyrovsky step in the Volmer-Heyrovsky mechanism can only be evaluated at potentials of $U = -0.3V_{SHE}$ and $-1.2V_{SHE}$. Therefore, it remains inconclusive whether the BEP relationship applies to the Heyrovsky step and we therefore omit this discussion in the manuscript.

Figure R6. (a) The BEP relationship associated with the Tafel step in the Volmer-Tafel mechanism on V_{20} - TiO_2 . (b) The BEP relationship associated with the Volmer step on V_{20} - TiO_2 and Au.

We have revised the text in the main manuscript in the middle of page 18, where we write as follows : “Surprisingly, the BEP relationship between reaction and activation energies holds..... predict electrocatalytic performance under the relevant reaction conditions”.

(8) The transition-state free energies of the Volmer and Heyrovsky steps in the HER are reduced to a different extent with increasing cathodic overpotential. The authors specified values of 1.01 eV and 1.10 eV in Figure 5c, respectively. This might indicate that at sufficiently large cathodic overpotentials, the Heyrovsky step will no longer limit the rate, but instead the Volmer step governs the electrocatalytic activity. I believe that this point should be discussed in more detail, which is possible by constructing free-energy diagrams along the reaction coordinate.

Our calculations suggest that the HER process on the V_{20} - TiO_2 surface follow to the Volmer-Tafel mechanism, see Figure R7a. Within this framework, the Volmer step proceeds spontaneously (see step ② in Figure R7), whereas the Tafel step emerges as the rate-determining step (RDS), see Figure R7b. As the energy barrier associated with the Tafel step is remarkably high, it needs a substantial cathodic overpotential to be operational. This result is consistent with experimental observations where larger HER current densities occur under large cathodic overpotentials (Figure 4a).

Figure R7. (a) HER process on V₂O₅-TiO₂. (b) The activation energy of Tafel step on V₂O₅-TiO₂.

(9) Ultimately, the construction of free-energy diagrams allows translating the calculated energetics to a Tafel plot, which can be directly compared to Figure 4. In case of the Volmer-Heyrovsky mechanism, the corresponding equations can be found in <https://doi.org/10.1002/elsa.202100037C>.

<https://chemistry-europe.onlinelibrary.wiley.com/doi/full/10.1002/elsa.202100037>

When the calculated transfer coefficient α exceeds 1, the standard Tafel slope formula ($(2.303RT)/(\beta nF)$, $\beta = 1 - \alpha$) is no longer applicable. In such cases, the relationship between electrode potential, electron transfer, and the HER mechanism on semiconductor electrodes is more complex than on metallic electrodes. This complexity arises from the simultaneous charge transfer in both polaron formation and the HER steps (see Figure R5). For more details, please refer to the response provided in question (6).

(10) “By increasing the potential to the experimental onset potential of -0.3 VSHE,” What is the ‘so-called’ onset potential? According to Compton, this term is ill-defined (<https://doi.org/10.1016/j.apmt.2019.05.011>), unless the chosen potential is clearly related to a current density, which is not the case in the discussion here.

To avoid ambiguity, we have modified this part to “By increasing the potential to -0.3 V_{SHE}”.

(11) It would be desirable if the grand canonical barriers were marked with a “#”, as this allows an easy distinction between the thermodynamic and kinetic quantities.

We have followed your advice and modified these symbols.

(12) Pages 17/18: the discussion of the RDS should be refined based on the comments above.

For this section, please refer to the response provided in question (4). Our results show that the rate-determining step (RDS) occurs at the Tafel step on the $V_{2O}-TiO_2$ surface.

(13) Page 18 “Impact of potential-dependent polaron formation on free energy relations, kinetics, and electrocatalyst design”: the discussion of the traditional Butler- Volmer formalism should be refined based on additional calculations and the comments above.

We removed the discussion on the Butler-Volmer formalism in the main manuscript and focused on analyzing the impact of polarons on reaction free energy, activation barrier and a transfer coefficient. We have included a short discussion on the validity of BV in the new section S10 on page S47-S48 in SI.

(14) The surface coverage of TiO_2 is dependent on the applied electrode potential. How do the authors ensure that they have chosen the ‘correct’ coverage under HER conditions for the modeling of the HER thermodynamics and kinetics?

I did not see that the authors determined surface coverages as a function of U. However, the omission of surface coverages can lead to erroneous results on the electrocatalytic activity (<https://doi.org/10.1039/D4CP01792G>). Therefore, I ask the authors to address this point seriously.

We fully agree that accounting for the surface coverage is crucial. However, in this case we have not carried out the complete mapping of the hydrogen coverage as a function potential, i.e. constructed Pourbaix diagrams, as this is extremely tedious and expensive with an explicit water model and as an explicit function of potential. We have, however, considered the adsorption of one and two protons on neighboring active sites as this is required for the Tafel step.

We find that on the TiO_2 surface, O atoms tend to easily interact with H atoms to form OH groups. Therefore, we examined the adsorption of a hydrogen atom on the oxygen atom nearest to the oxygen vacancy, leading to the formation of an OH group, due to its direct and significant influence on the activity of oxygen vacancy, see step ① in Figure R4a. Please refer to the more detailed response provided in question (4).

Reviewer #2

In this manuscript, Honkala and co-workers reported potential-dependent polaron formation and its impact on electrocatalytic HER with TiO_2 semiconductor electrodes. The detailed theoretical calculations and experimental results well support their view. The results are interesting and the data presented are ample. Therefore, this work can be considered for publication in Nature Communications after some revisions.

We appreciate your positive feedback on our work. We also thank you for the valuable comments, which we answer in detail below.

(1) The author's description of the order of the pictures is very confusing, making it difficult to read the corresponding content smoothly. Please reorganize the order of the pictures and

texts for reading easily. All the data in Supporting Information should be mentioned and labeled in the Manuscript.

We appreciate the referee's suggestion to cite all data (tables and figures) included in the Supporting Information within the main manuscript. While we agree that referencing key supplementary materials improves clarity and accessibility, citing *every* item is unfortunately not feasible due to the large volume of supporting data and the need to maintain readability and flow in the main text.

Nonetheless, we have ensured that all critical data supporting our main conclusions are now appropriately cited and discussed in the manuscript. We have also revised the Supporting Information to include clearer labeling and cross-references, where helpful.

(2) The author mentioned the corresponding charge value changes when describing the DOS in Figure 1. Based on the reviewer's understanding, only the DOS can qualitatively determine the electron distribution position. Please explain how to determine the specific value of the change of corresponding charges, and suggest that it be clearly marked in figures.

While the primary role of DOS is to quantitatively describe the distribution of electronic energy levels, integrating the DOS over the given energy range ($\int_{E_1}^{E_2} \rho(\epsilon) d\epsilon$) provides the number of states between energies E_1 and E_2 . However, this approach does not provide quantitative information about the spatial distribution of electrons. By correlating the atomic spin magnetic moment and DOS, it is possible to effectively capture both the spatial localization and the energy level of e.g. a polaronic state. This method is particularly applicable to non-magnetic TiO₂ systems where the quantitative relationship between magnetic moment and electron localization is clear. This is due to the single source of magnetic moment and the absence of spin coupling interference. The atomic magnetic moments in TiO₂ arise from excess electrons forming polarons so the DOS and atom spin magnetic moments together quantitatively describe electron localization and energy.

Please see Section S1 on page 3 in the SI for using DOS to analyze polaron formation.

(3) The reviewer noticed that the author mentioned that the AMM of Ti_{5c}³⁺ is stable at U=-2 V, but there were still several shaking points, and as increased to U=-3 V, the shaking became more intense. How to explain the cause of the shaking points? Or is there a charge transfer in the kinetic process?

By monitoring the AMM of Ti_{5c}³⁺ as a function of time, we observe that Ti_{5c}³⁺ remains stable but exhibits several sudden fluctuations at $U = -2.0 V_{SHE}$, as indicated by the black arrows in Figure R1a. We would like to note the spikes are rather small in intensity, ~ 0.1 AMM, and likely result from small fluctuations in the atomic positions and system charge during the constant temperature

– constant potential molecular dynamics. We also analyzed the time-dependent surface charge during the CIP-DFT-MD simulation at $U = -2.0 \text{ V}_{\text{SHE}}$ (Figure 8a) and found that the excess electrons display small thermal fluctuations around the average number of excess electrons (Figure R8b). These fluctuations are normal and expected for any system studied using constant potential MD as the charge fluctuations result from the action of the (computational) potentiostat fixing the electrode potential. See for instance the original CIP-DFT paper (<https://www.nature.com/articles/s41524-023-01184-4>).

Figure R8. (a) The variation of AMM for a Ti_{5c}^{3+} polaron state on TiO_2 (101) at $U = -2.0 \text{ V}_{\text{SHE}}$. The arrows indicate where the AMM value significantly decreases when time varies. (b) The comparison of excess electron variation as a function of time at $U = -2.0 \text{ V}_{\text{SHE}}$.

The above discussion is given on page 13 in SI and Figure R8 is presented as Figure S8 in the SI.

(4)The CIP-DFT method is a very important research method for analyzing electrochemical systems. The reviewer suggested that the key points of the CIP-DFT method be briefly described in the calculation details of the main text, which is conducive to its promotion and application.

We have revised the description of the CIP-DFT method, **please see the computational details section in the main manuscript.**

(5) The O 1s and Ti 2p XPS spectra should be provided.

The high-resolution XPS analysis of the Ti 2p region ((see Figure R9a) exhibits peaks at 458.5 and 464.3 eV, which correspond to the Ti $2p_{3/2}$ and Ti $2p_{1/2}$ states in pristine TiO_2 , respectively. Following the incorporation of V_O defects, the binding energies associated with Ti $2p_{3/2}$ and Ti $2p_{1/2}$ show negative shifts of 0.29 and 0.36 eV, respectively, indicating the presence of low-valence Ti species. Additionally, the O 1s spectra (Figure 9b) reveal a notable increase in the V_O peak area for V_O - TiO_2 relative to pristine TiO_2 , implying an elevated concentration of V_O defects (Chem. Catal. 2021, 1, 1437-1448).

Figure R9. XPS spectra of TiO₂ and Vo-TiO_{2x} in (f) Ti 2p and (g) O 1s regions.

Figure 9 and the above discussion can be found from page 39 in the SI.

(6) CV tests need to be performed.

We conducted cyclic voltammetry (CV) tests and observed that, compared with pristine TiO₂, Vo-TiO₂ exhibits distinct oxidation peaks corresponding to the transition from Ti³⁺ to Ti⁴⁺ and reduction peaks corresponding to the transition from Ti⁴⁺ back to Ti³⁺, see Figure R10. These findings indicate that the formation of Ti³⁺ polarons in Vo-TiO₂ is potential-dependent and reversible.

Figure R10. Cyclic voltammetry (CV) tests of TiO₂ and Vo-TiO₂ systems.

Figure R10 is now presented as Figure S29 in SI and the above text can be found from page 40 in SI and briefly mentioned on page 12-13 in the main manuscript.

(7) In Table S9, there is no blue or green text.

This mistake has now been corrected and the colours have been added, see page 46-47 in the SI.

Reviewer #3

In this article, the authors studied the formation and change of the Ti^{3+} and oxygen vacancy on TiO_2 in the (photo)electrocatalytic process using in situ experiments. Despite the wealth of experimental and theoretical calculations, the novelty of the conclusions and introduction regarding Ti^{3+} and oxygen vacancies are insufficient, and a large experimental literature already exists. We therefore recommend that the authors transfer to a more specialized journal.

While several studies have explored the influence of polarons on catalytic activity, to the best of our knowledge, our work is the first to explicitly highlight the critical role of the electrode potential in polaron formation and its impact on activating the HER reaction. Reaching this conclusion required the integration of state-of-the-art simulation and experimental techniques. Specifically, this is the first study to employ the novel CIP-DFT method, or any computational, method, alongside experiments to demonstrate the formation and importance of *potential-dependent* polarons on electrochemical processes at the atomic scale. Furthermore, other reviewers have recognized the originality of our results.

We have modified the manuscript to highlight the novel aspects of our work:

- 1) We show that the TiO_2 electronic structure can be substantially controlled by the application of an electrode potential due to the potential-dependent reduction of Ti^{4+} to Ti^{3+} surface polarons. (Figure 1)
- 2) We examine the stability and dynamics of the Ti^{3+} polarons using advanced constant inner potential (CIP-DFT) molecular dynamics (MD) and demonstrate that a single Ti^{3+} polaron is relatively stable and capable of hopping between Ti atoms via electron transfer while bipolaron is unstable at room temperature. (Figure 2 and Figure 3a-f)
- 3) Through CIP-DFT studies we show that potential-dependent polaron formation breaks down the linear CHE-like relationship between adsorption energy and electrode potential; this highlights that the potential-dependent activity of semiconductor electrodes differs fundamentally from that of metals (Figure 6a in the manuscript).
- 4) We show that the simultaneous charge transfer associated with both polaron formation and the HER in TiO_2 causes the transfer coefficient α to exceed 1, indicating that the HER and charge transfer mechanisms on semiconductor electrodes is more complex than on metallic electrodes. We also indicate that the activation energy of the chemical Tafel step depends on the applied electrode potential even though it remains independent of the electrode potential in metallic systems (Section Impact of potential-dependent polaron formation on free energy relations, kinetics, and electrocatalyst design).

Reviewer #4

The authors, from a strong collaboration among different theoretical groups, report a thorough study on potential-dependent changes of semiconducting electrodes. While electrode/catalyst surface restructuring under working conditions is actively studied experimentally, their atomic structure remains unclear. This paper, focused on density functional theory-based calculations, provides noteworthy results of significance to the field. However, the involved experimental work and its link with the theoretical results need further development to fully support the conclusions. To substantiate the novelty of this work as stated in the abstract “combining calculations with multiple *in situ* experiments”, major revisions are required before publication, especially for the broad audience of the journal.

We thank the reviewer for their valuable comments and response to each of them in detail below.

(1) In the *in-situ* UV-Vis spectra (Fig. 2d), the absorbance peak around 630 nm is poorly defined at -2.0 V_{SHE} but more pronounced at -2.2 V_{SHE}. Why the calculation (Fig. 2a) was not carried out at -2.2 V_{SHE} should be justified. Since 630 nm corresponds to 1.97 eV, should the label in Fig. 2d “Ti³⁺ polaron” be changed to “Ti_{5C}³⁺” to link to Fig. 2a? 可以 In the *in-situ* EPR spectra (Fig. 2e), no experimental data is provided at potentials more negative than -2.4 V_{SHE}. How does this relate to the calculation (Fig. 2c) at -3.0 V_{SHE}? In the *in-situ* EFM maps (Fig. 2f-h), potentials are measured in solid-state (dry) conditions. How do they relate to the electrochemical (wet) conditions used in the calculations (Fig. 2a-c)? Referring to Fig. 1, it is understandable that -2.0 and -3.0 V_{SHE} were chosen from modelling, but experimental support is insufficient in Fig. 2 especially when potential is key to this work.

To avoid confusion and to clarify our discussion, we removed the *in-situ* UV-Vis spectra from the manuscript, because a non-aqueous organic electrolyte was used to measure it. The actual electrocatalytic HER studies were conducted in an aqueous electrolyte and therefore the UV-Vis spectra might not accurately represent the true *in situ* conditions of HER. We have also removed the *in situ* EFM maps as they were measured under atmospheric conditions rather than the aqueous HER conditions and because their low resolution cannot provide very intuitive atomic-scale information needed in this work. Additionally, the surface potential measurements in liquid present inherent challenges, primarily due to the shielding effect exerted by free ions in the solution on the tip-sample voltage, which fundamentally impedes accurate surface potential determination through capacitive force measurements. Therefore, we consider that the *in situ* EFM methodology is not suitable for the purposes of the present study. Both the UV-Vis and EFM measurements are replaced with *in situ* EPR spectra recorded in the aqueous electrolyte as they were observed to be more accurate and insightful.

In the *in situ* EPR spectra, we discerned that the spectral lines started to exhibit subtle fluctuations after -2.0 V_{SHE}, with the emergence of a distinct Ti³⁺ signal peak at -2.4 V_{SHE}, see Figure R11a. Furthermore, this signal underwent a remarkable amplification at the more negative potential of -3.0 V_{SHE}. Consequently, we addressed the stability and dynamics of the Ti³⁺ polarons as a function of the electrode potential through CIP-DFT molecular dynamics (MD) simulations. Overall,

the Ti^{3+} polaron formation potentials from *in situ* electrochemical EPR measurements and CIP-DFT calculations show excellent agreement, with a potential deviation of only $0.1 \text{ V}_{\text{SHE}}$. Figure R11b shows that a Ti^{3+} polaron localizes on the Ti_{6c} site upon structure optimization at $U = -2.0 \text{ V}_{\text{SHE}}$. However, in the CIP-DFT-MD simulations the polaron instantaneously moves from a Ti_{6c} site to a Ti_{5c} surface site due to thermal motion and this change is accompanied by a small energy penalty (see the energy difference (ΔE) in Figure R11b) which is counterbalanced by an entropy gain from the availability to more localization sites. Monitoring the AMM of Ti_{5c}^{3+} over time reveals that it is stable and exhibits only minor fluctuations, as indicated by black arrows in Figure R11c.

At more reducing potential of $U = -2.4 \text{ V}_{\text{SHE}}$, a bipolaron can intermittently localize on either the of two neighboring Ti atoms or the next-nearest neighboring Ti atoms with frequent transitions between these configurations, see Figure R11d. The large oscillations in AMM values shown in Figure R11d suggests that a localized bipolaron state may be unstable at room temperature. When the potential is further increased to $-3.0 \text{ V}_{\text{SHE}}$, the significant oscillations in AMM values persist, see Figure R11e. The difference between -3.0 V and -2.4 V is the increased oscillation frequency of AMM values under more negative potentials, suggesting a gradual destabilization of the bipolaron state.

Figure R11. (a) In situ EPR spectroelectrochemistry of a TiO_2 -modified ITO electrode in $0.1 \text{ M H}_2\text{SO}_4$ electrolyte. (b) The dynamic behavior of Ti^{3+} polaron state on TiO_2 (101) calculated with the CIP-DFT-MD at $U = -2.0 \text{ V}_{\text{SHE}}$. The plot shows that during CIP-DFT-MD the polaron is localized on the (Ti_{5c}) Ti atom. The dashed line represents the energy of Ti_{6c}^{3+} cation calculated with the static CIP-DFT approach at $U = -2.0 \text{ V}_{\text{SHE}}$. (c) The variation of AMM for a Ti_{5c}^{3+} polaron state on TiO_2 (101) at $U = -2.0 \text{ V}_{\text{SHE}}$. The arrows indicate where the AMM value significantly decreases when time varies. (d) The variation of AMM for $\text{Ti}_{5c}^{3+}/\text{Ti}_{5c}^{(4-\delta)+}$ polaron states on TiO_2 (101) at $U = -2.4 \text{ V}_{\text{SHE}}$. δ in the superscript indicates excess electrons ($0 < \delta \leq 1$). (e) The variation of AMM for $\text{Ti}_{5c}^{3+}/\text{Ti}_{5c}^{(4-\delta)+}$ polaron states on TiO_2 (101) at $U = -3.0 \text{ V}_{\text{SHE}}$.

Figure R11 and above text can be found from the main manuscript: Figure R11 is presented in Figure 2 and text highlighted in blue on page 9 in the main manuscript.

(2) The disconnection between experiments and theory is also evident in the SI. In the in-situ EXAFS spectra (Fig. S9), how does the Ti-O bond change relate to the calculations (Fig. 2a-

c)? As the main claim of this paper is that the applied potential alters the electronic state of the electrode surface, in-situ Raman spectroscopies of TiO₂ should be potential-dependent (see examples in *Energy Environ. Mater.* 2024, 7, e12692 and *ACS Appl. Nano Mater.* 2023, 6, 6528). The unchanged spectra (Fig. S37) contradict this claim. They should be enlarged to show intensity and position changes with potential.

The EXAFS analysis in Figure S11 (Figure R12) demonstrates a decrease in the intensity of the Ti-O bond peak from OCP to -2.0 V vs SHE, indicating local lattice distortions. Additionally, the peak shifts to longer distances suggest an increase in the Ti-O bond length with more negative electrode potentials (Figure R12a). EXAFS fitting results for the Ti-O bond length confirm this trend, which also aligns with the CIP-DFT calculation results (Table R1). Notably, both experimental and computational data show a clear correlation between increasingly negative potentials and Ti-O bond elongation, driven by potential-dependent Ti³⁺ polaron formation in TiO₂. However, a minor discrepancy (~0.05 Å) in average bond length values is observed, but it remains within the acceptable error margin (<5%). This variation is likely due to environmental factors in experimental measurements and approximations in theoretical simulations.

Figure R12. (a) In situ EXAFS spectra of Ti R space for V_O-TiO₂. (b-d) Corresponding FT-EXAFS fitting curves.

Table R1. EXAFS fitting parameters at the Fe K-edge. Bond distance unit: (Å)

Potential	Path	Coordination number	Bond distance (experiment)	Bond distance (DFT)
OCP	Ti-O	5.1±1.2	1.951±0.051	1.991
-1.0V _{SHE}	Ti-O	5.0±0.9	1.953±0.016	2.003
-2.0V _{SHE}	Ti-O	4.8±1.0	1.956±0.017	2.009

Figure R13. The main peak of pristine TiO₂ in the in situ Raman spectroscopies.

The *in situ* Raman spectroscopy results reveal that the main Raman signal of TiO₂ undergoes significant attenuation under high potential conditions, see Figure R13. This behavior can be ascribed to the formation of oxygen vacancies or local bond length distortions within the lattice, which disrupt the symmetry of vibrational modes and consequently reduce the Raman scattering efficiency. Notably, the Raman peak position exhibits continuous fluctuations under high potential conditions, likely associated with the potential-dependent polaron formation.

Figures R12 and R13 and above discussion has been included into SI on pages 16 and 41, receptively. We also provided a detailed introduction in the main text, see pages 12-14.

(3)In the introduction, an overview of the latest theoretical methods and results from the established literature is missing. This makes it difficult to compare the theoretical work with the state-of-the-art. The current overview of the experimental methods should be compressed to enhance relevance to the calculations. The introduction to the polaron and defects, as the

focus of this work, is scattered across pages 4 and 5 and should be rearranged and brought forward.

We now emphasize the latest theoretical methods for semiconductor electrocatalysis, see page 3 and 4 in the main manuscript. Moreover, we also introduce polarons on page 3.

(4) In the methods, more experimental details should be provided to ensure reproducibility. For example, the composition of the Ag/AgCl electrode, as the reference electrode, changes depending on whether 1M, 3M, or saturated KCl is used. The manufacture and model of electrodes and cells should be specified; if home-made, photos or schematics with parameters should be provided in the SI. This is crucial as in-situ measurements are often restricted by limited space (e.g., EPR). Furthermore, the laser power for Raman spectroscopy should be provided.

We have provided more experimental details in the main manuscript as suggested.

The commercial reference electrode used is a 3 M KCl Ag/AgCl electrode, as specified in the original manuscript. Device designs for in situ electrochemical EPR and XAFS, are given in Figure R4 and Figure R5. *In situ* Raman measurements were carried out on an XploRA PLUS Raman spectrometer equipped with a 50× objective and a 638 nm He-Ne laser.

[FIGURE REDACTED]

Figure R14. In situ electrochemical EPR device design. This was originally presented in a published article from our group [Nat. Synth. 3, 763-773 (2024)].

Figure R55. In situ XAFS device design.

Figures R14 and R15 have been added into SI on pages 41 and 42, respectively.

(5) Throughout the manuscript, abbreviations should be defined when they first appear, e.g., V_O -TiO₂(101).

V_O stands for oxygen vacancy (V_O) formation and was defined in page 5 in the introduction. We have checked that all abbreviations are defined when first used.

Reviewer 5.

In the revised manuscript, the authors have introduced in-situ EPR spectroelectrochemistry, in-situ XAS, and in-situ Raman analyses to support their claim regarding the formation of surface Ti_{5c}^{3+} polarons during the reaction. However, these additions raise several critical concerns that might significantly impact the conclusions and overall claims of this manuscript. The following issues must be addressed before the manuscript can be considered for publication:

We thank the reviewer for careful consideration of our work. Below we address in detail all raised issues and demonstrate that surface polarons form under the reaction conditions.

1. The in-situ EPR spectroelectrochemistry conducted on TiO_2 -modified ITO electrode in 0.1 M H_2SO_4 electrolyte demonstrates the ability to detect surface Ti_{5c}^{3+} species at room temperature, suggesting their lifetime ranges from milliseconds to seconds. In contrast, the CIP-DFT-MD simulations indicate a much shorter timescale in the picosecond range. This significant mismatch must be reconciled, and the applicability of the experimental and theoretical approaches to each other should be clearly explained.

We fully agree that EPR is a macroscopic technique and detects the polaron formation signals in the millisecond-second range while the CIP-DFT-MD simulations demonstrate the polaron formation on much shorter time/length scale. There are, however, no mismatch between the computational and experimental results: there is no indication in the CIP-DFT-MD results that the polaron would not be dynamically stable beyond the picosecond range. Even longer 10ns simulations obtained with state-of-the-art machine learning models show that in bulk anatase electron polarons are stable [<https://doi.org/10.1103/PhysRevLett.134.216301>]. Furthermore, electron polarons in TiO_2 are known to be more stable on the anatase surface than in bulk [*J. Phys. Chem. C* 2018, 122, 27540-27553.; *Phys. Status Solidi RRL* 2014, 8, 583-586.] Combined with our results, these previous studies show that surface polarons are (thermo)dynamically stable at least in the nanosecond timescale.

In practice, it is technically or computationally impossible to perform CIP-DFT-MD or any other DFT-MD simulations that reach the requested millisecond timescale. To our knowledge, no DFT-MD studies have achieved even the microsecond timescale. The CIP-DFT-MD is approximately 20-30% more computationally demanding than standard DFT-MD, making the requested millisecond timescale even less attainable. While machine learning potentials can extend DFT-quality MD simulations into the nanosecond or microsecond range

for some metallic systems, they are currently unable to simulate complex semiconducting oxide – water interfaces under constant potential conditions. Even for bulk TiO₂, state-of-the-art machine learning models achieve 10 nanosecond simulations [<https://doi.org/10.1103/PhysRevLett.134.216301>]. Achieving MD simulations on the millisecond to second timescale even with the classical potentials is not feasible for TiO₂ – water interfaces.

We would also like to highlight that the CIP-DFT and CIP-DFT-MD developed by us (*npj Computational Materials* 10, 5 (2024)) are themselves highly novel. Currently, CIP-DFT-MD is the only available method capable to directly simulating and studying the impact of potential-dependent polaron formation in semiconductors. The present work is the first successful application of a potential-dependent DFT-MD framework to a semiconductor-water interface. Our simulations demonstrate that the CIP-DFT and CIP-DFT-MD methods can effectively capture the formation potential of Ti³⁺ polarons (see below) and provide new insights into their dynamic behavior within the picosecond time range - a phenomenon not previously reported in literature.

We believe it is not necessary to reach the millisecond timescale to demonstrate the Ti³⁺ polaron formation. Instead, polaron formation potential serves as reliable metric for assessing potential-dependent polaron generation. The time-independent CIP-DFT calculations indicate that the Ti³⁺ surface polarons are thermodynamically stable under reductive conditions. Complementary CIP-DFT-MD simulations further confirm their dynamic stability at room temperature and under the same reductive environment (Figure R1). Experimentally, the formation of stable surface polarons is corroborated by in-situ EPR experimental techniques. Notably, in-situ electrochemical EPR enables the identification of the electrode potential at which Ti³⁺ polarons are generated. The polaron formation potentials derived from EPR and CIP-DFT exhibit excellent agreement, with a minimal deviation of only 0.1 V_{SHE}.

In response to the reviewer' comment, we extended the simulation timescale and observed that a single potential-dependent Ti³⁺ polaron remains stable over an extended period, whereas multiple coexisting Ti³⁺ polarons are dynamically unstable. Please see Figure R1. These findings do not change the original conclusions. Although running the CIP-DFT-MD simulations at longer timescales remains a significant challenge, our team is actively exploring potential solutions. However, this is a huge project and lies beyond the scope of the current work.

Figure R1. (a) The variation of AMM for a Ti_{5c}^{3+} polaron state on TiO_2 (101) at $U = -2.0 \text{ V}_{\text{SHE}}$. The arrows indicate where the AMM value exhibits most notable fluctuations. (b) The variation in AMM for $\text{Ti}_{5c}^{3+}/\text{Ti}_{5c}^{(4-\delta)+}$ polaron states on TiO_2 (101) at $U = -2.4 \text{ V}_{\text{SHE}}$. δ in the superscript indicates the number of excess electrons ($0 < \delta \leq 1$). The yellow shaded area indicates the unexpanded time scale.

Modifications: Figure R1 and above discussion has been added into the in Supporting Information on pages 14 and highlighted in blue.

2. If the presence of surface Ti_{5c}^{3+} polarons during reaction is to be substantiated, surface-sensitive in-situ techniques—such as soft XAS (Ti L-edge) or in-situ XPS—should be utilized. The current reliance on Ti K-edge XAS, a bulk-sensitive method, is insufficient to confirm the formation of surface-localized species.

First, it should be noted that numerous prior studies have indicated that Ti^{3+} polarons are more likely to form on the surface of TiO_2 compared to within the bulk of the material. [John J. Carey, et al. Does Polaronic Self-Trapping Occur at Anatase TiO_2 Surfaces? *J. Phys. Chem. C* 2018, 122, 27540-27553.; Peter Deák, et al. Polarons and oxygen vacancies at the surface of anatase TiO_2 , *Phys. Status Solidi RRL* 2014, 8, 583-586.] We also agree that both Ti L-edge spectroscopy and X-ray photoelectron spectroscopy (XPS) are classified as soft X-ray techniques, which can achieve the need surface resolution missing from hard X-rays. However,

the photoelectron signals generated by soft X-rays are highly vulnerable to attenuation and distortion under atmospheric conditions and therefore achieving high-quality *in-situ* soft X-ray absorption spectroscopy (XAS) at the Ti L-edge and *in-situ* XPS measurements under atmospheric conditions remains technically challenging. Given these technical challenges, we were unable to obtain high-quality XAS or XPS measurements at the Ti L-edge.

As the central issue raised by the reviewer is to provide evidence for the presence of Ti^{3+} polarons on the TiO_2 surface, we verify the formation of Ti^{3+} surface polarons *in-situ* electrostatic force microscopy (EFM) measurements under atmospheric conditions. This measurement method is highly sensitive to surface charges and can detect the formation of polarons induced by the potential. The EFM topography image in Figure R2a displays a uniform profile of the sample surface after being subjected to a range of tip biases from -1 V to -7 V and suggests that sample has not undergone structural deformations. The EFM 1ω signal mapping in Figure R2b shows that when the TiO_2 sample is exposed to negative tip biases, a noticeable surface charge appears. When the applied voltage is further decreased from -1 V to -2 V, the 1ω signal increases, illustrating the potential-dependent local charge accumulation on the surface, see Figure R2b. We ascribe the observed EFM signals to the aggregation of Ti^{3+} polarons on the surface. This assignment is in line with the absence of pronounced features in the 2ω signal, suggesting that the dielectric constant and thus the stoichiometry of the TiO_2 sample remains unchanged under different electrode potential conditions, see Figure 2c. It is also important to note that the *in-situ* EFM maps were obtained under atmospheric conditions rather than aqueous conditions. This is because measuring surface potentials in liquid phases poses significant challenges, primarily due to the shielding effect of free ions in the solution on the tip-sample voltage. This shielding fundamentally limits the accuracy of surface potential determination via capacitive force measurements. Even though the *in-situ* EFM maps were not obtained under aqueous conditions, this methodology remains a reliable approach to demonstrate the significant tendency of Ti^{3+} accumulation on the TiO_2 surface.

In addition to the EFM measurements of the TiO_2 sample at -3.0 V, the sample was further characterized using X-ray absorption near-edge structure (XANES) spectroscopy in the total electron yield (TEY) detection mode. The TEY mode is surface-sensitive as it primarily detects secondary electrons emitted from the catalyst surface, with an electron escape depth of approximately 4 nm [*Chem. Mater.* 2016, 28, 12, 4467–4475.]. The TEY XANES spectra of the TiO_2 sample at the Ti $L_{3,2}$ -edge are presented in Figure R3. Typically, the Ti $L_{3,2}$ -edge probes electronic transitions from the Ti 2p core levels to unoccupied states of Ti 3d character. This includes two pre-edge features (peaks m and n), which can be attributed to core-hole–d-electron interactions. The Ti L_3 -edge (peaks a–c) and L_2 -edge (peaks d and e) correspond to electronic transitions from the spin-orbit split Ti $2p_{3/2}$ and $2p_{1/2}$ initial states, respectively, to the Ti 3d final states. The splitting into distinct t_{2g} and e_g peaks within both the Ti L_3 -edge and L_2 -edge arises due to crystal field effects. Most importantly, the additional splitting of the e_g peak (into peaks b and c) at the Ti L_3 -edge is indicative of local tetragonal distortion at the Ti site

within the TiO_6 octahedron. Furthermore, the degree of e_g peak splitting correlates with the oxidation state of Ti. Experimentally, the reduced resolution of peaks b and c in TiO_2 after applying -3.0 V suggests a lower oxidation state of Ti, implying the presence of some $\text{Ti } 3d^1$ (Ti^{3+}) character. [*Chem. Mater.* 2016, 28, 12, 4467–4475.] Furthermore, the L_3 and L_2 edges specifically correspond to electrons in the spin-up (spin quantum number $m_s = +1/2$) and spin-down ($m_s = -1/2$) states, respectively. After applying -3.0V , TiO_2 exhibits peaks associated with electron spin polarization that have greater intensity than those of pristine TiO_2 , indicating a higher proportion of electrons in spin-polarized Ti^{3+} states. [*ACS Catal.* 2024, 14, 1, 249-261.]

Overall, previous calculations [*J. Phys. Chem. C* 2018, 122, 27540-27553, *Phys. Status Solidi RRL* 2014, 8, 583-586] as well as our EFM and TEY-XANES experiments show that surface Ti^{3+} form under the reaction conditions.

Figure R2. (a-c) Topography maps (a), EFM- 1ω (b) and EFM- 2ω (c) measurements on the pristine TiO_2 sample subjected to tip biases in a range of $-1 \sim -7\text{V}$ at different locations. Scale bar, $2 \mu\text{m}$. The regions of interest are marked by circles.

Figure R3. Ti $L_{3,2}$ -edge XANES spectra of TiO_2 before and after applying voltage.

Modifications: Figure R2, R3, and the above discussion have been added into the Supporting Information on pages 15-17 and is highlighted in blue.

3. The Ti K-edge XAS data show negligible shifts in the absorption edge and EXAFS features, which do not convincingly support the presence of surface Ti_{5c}^{3+} polaron. This stands in contrast to the significant EPR signal, which implies a considerable population of such surface states, which may be detected. The inconsistency between the two techniques must be resolved.

To address this point, the TiO_2 and V_O - TiO_2 samples were subjected to in-situ electrochemical spectroscopic analysis for comparative evaluation. Unlike TiO_2 , V_O - TiO_2 inherently contains a significant concentration of Ti^{3+} , which results in an absorption edge shift of 0.37 eV in the K-edge XANES, see Figure R4. When a potential ranging from OCP to -2.0 V_{SHE} is applied, the absorption edge undergoes an additional displacement of 0.23 eV, accounting for 62% of the total shift observed between TiO_2 and V_O - TiO_2 , see Figure R4. These findings strongly suggest that the presence of Ti^{3+} shifts the K-edge XANES peak by ~0.2-0.3 eV depending on the potential (from OCP to -2.0 V_{SHE}).

Furthermore, the EXAFS analysis reveals a significant reduction in the Ti-O coordination number (CN) in V_O - TiO_2 compared to pure TiO_2 through coordination number fitting, see Figure R5 and Table R1. Upon applying a potential, it is observed that the Ti-O coordination number gradually decreases with the application of negative potential; this again suggests that the formation of Ti^{3+} during the potential-dependent process is accompanied by the generation of a small number of oxygen vacancies. This result is further corroborated by the CIP-DFT calculations of V_O formation energies at various potentials, see Table R2. It can be seen that as

the negative electrode potential increases from 0 V_{SHE} to $-2.0 V_{\text{SHE}}$, the negative surface charge of $\text{TiO}_2(101)$ increases linearly and the oxygen vacancy formation energies decrease linearly. This accumulated surface charge is directly associated with the formation of Ti^{3+} polarons, as explicitly illustrated in Figure 1 of the main text, thereby leading to a decrease in oxygen vacancy formation energies. The facilitated oxygen vacancy formation at more reducing potentials can be attributed to the fact that the negatively charged $\text{V}_\text{O}\text{-TiO}_2(101)$ surface gets stabilized by the positive charge left behind by the oxygen, which brings the slab closer to a charge neutral state. It should be noted that polarons are localized mainly in the surface region while both K-edge XANES and EXAFS probe both the surface and bulk regions of the material and are not surface-sensitive techniques, as discussed below in point 4. Therefore, the fitting parameters and e.g., the small changes in the average bond lengths due to polaron formation do not align with the CIP-DFT results, which specifically represent only the surface region and the effects of polaron formation.

Overall, both the K-edge XANES and EXAFS support the formation of Ti^{3+} polarons, which localize on the surface region as discussed in our response to point 2.

Figure R4. (a) Ti K-edge XANES spectra of different TiO_2 -based samples. (b) Comparison of normalized Ti K-edge absorption edge energy positions.

Figure R5. EXAFS spectra of Ti R space for different TiO₂-based samples.

Table R1. EXAFS fitting parameters at the Ti K-edge.

Potential	R-factor	Path	C.N.	R (Å)	ΔE_0 (eV)	σ^2 (Å ²)
TiO ₂	0.019	Ti-O	6*	1.928±0.012		0.005
Vo-TiO ₂	0.014	Ti-O	5.2±0.5	1.934±0.021		0.005
OCP	0.013	Ti-O	5.2±0.6	1.942±0.010	-4.86	0.005
-1.0V _{SHE}	0.015	Ti-O	5.0±0.7	1.944±0.011		0.004
-2.0V _{SHE}	0.014	Ti-O	4.8±0.7	1.946±0.015		0.003

Table R2. The formation energies of an oxygen vacancy (E_{Vo}) on TiO₂(101) at PZC and different electrode potentials at the GGA+U level.

Electrode potential (V _{SHE})	E_{Vo} (unit:eV)	Charge(e ⁻)
PZC	2.97	0
0	2.73	-0.26
-0.5	2.56	-0.67
-1.0	1.87	-1.04
-2	0.68	-1.81

Modifications: Figure R4 has been added into the main manuscript in Figure 3h. Figure R5 has been added into the Supporting Information on pages 20 and 21, and the accompanying

text is given in blue. Table R1 and R2 have been added into the Supporting Information on pages 21 and 41, and the related text is given in blue.

4. A comparative analysis of the Ti K-edge XAS spectra for pristine TiO₂ and VO-TiO₂ should be provided to clarify any differences in Ti oxidation states and local environments. This comparison is essential for substantiating claims regarding Ti state modulation.

To clarify this issue, we performed a detailed comparison of the Ti K-edge XAS spectra for pristine TiO₂ and V_O-TiO₂, see Figure R6. Our findings show that after vacancy formation, the absorption edge moves toward lower oxidation states of Ti, see Figure R6a.

The EXAFS analysis, based on coordination number fitting (see Figure R6b and Table R3), reveals a significant reduction in the Ti-O coordination number in V_O-TiO₂ compared to pure TiO₂. However, no significant differences were observed in the average Ti-O bond lengths between these two systems, see Table R3. The minor impact of O-vacancies on the average Ti-O bond length can be attributed to the nature of EXAFS, which provides macroscopic, average structural information of materials. As a result, local changes in bond lengths may be masked and not clearly reflected at the macroscopic level.

To overcome this limitation, the CIP-DFT method employed in this study enables atomic-scale insight into potential-dependent local structural changes due to polaron formation and thereby overcomes some limitations of the experimentally obtained macroscopic averaged data. The CIP-DFT results demonstrate that the formation of Ti³⁺ under varying potential leads to a measurable elongation of Ti-O bonds from 1.991 to 2.009 Å and thereby show that the local changes in the microscopic structure are far greater than in the macroscopic experiments.

Figure R6. (a) Comparison of normalized Ti K-edge absorption edge energy positions for different TiO₂-based samples. (b) EXAFS spectra of Ti R space for different TiO₂-based samples.

Table R3. EXAFS fitting parameters at the Ti K-edge.

Potential	R-factor	Path	C.N.	R (Å)	ΔE_0 (keV)	σ^2 (Å ²)
TiO ₂	0.019	Ti-O	6*	1.928±0.012	-4.86	0.005
Vo-TiO ₂	0.014	Ti-O	5.2±0.5	1.934±0.021		0.005

C.N., σ^2 , R-factor represented the coordination number, Debye-Waller factor, and goodness of the fit, respectively. R referred to the bond distance. The energy shift (ΔE_0) was set for all spectra to highlight the variation of R. S_0^2 was set to 0.68, according to the experimental EXAFS fitting of TiO₂ reference by fixing CN as the known crystallographic value.

Modifications: Figure R6 has been added into the Supporting Information on page 19 and the related text is given in blue.

5. The in-situ XAS measurements were conducted at -1.0 and -2.0 VSHE, yet in-situ EPR data suggest no significant Ti state variation until -2.4 VSHE. To enable meaningful correlation between the two datasets, XAS measurements should be performed at potentials consistent with those used in EPR analysis.

In the case of the pristine TiO₂ electrode, in-situ XAS analysis was not conducted due to significant hydrogen bubble formation on the catalyst surface at highly negative potentials (below -2.0 V_{SHE}). This bubble formation in the aqueous phase severely interferes with the detection of the low-energy Ti K-edge around 4.86 keV. As a result, in-situ EPR measurements were employed instead to investigate the pristine TiO₂ system, where Ti³⁺ formation occurs exclusively under highly negative potentials. Therefore, in-situ EPR is the most suitable method for characterizing the formation potential of Ti³⁺ in this study.

The presence of oxygen vacancies in Vo-TiO₂ enhances its conductivity, allowing the potential-dependent formation of Ti³⁺ polarons to in TiO₂ be studied at comparatively less negative potentials. Accordingly, in-situ XAS measurements were conducted from OCP to -2.0 V_{SHE} to monitor structural changes associated with Ti³⁺ polaron formation. These measurements revealed a slight decrease in the coordination number of Ti-O with increasing negative potential, along with a minor elongation of the Ti-O bond, see Table R4. For further discussion of the expected bond length changes observed in the XAS spectra, please refer to the discussion in point 4.

Table R4. EXAFS fitting parameters at the Ti K-edge.

Potential	R-factor	Path	C.N.	R (Å)	ΔE_0 (eV)	σ^2 (Å ²)
TiO ₂	0.019	Ti-O	6*	1.928±0.012		0.005
V _o -TiO ₂	0.014	Ti-O	5.2±0.5	1.934±0.021		0.005
OCP	0.013	Ti-O	5.2±0.6	1.942±0.010	-4.86	0.005
-1.0V _{SHE}	0.015	Ti-O	5.0±0.7	1.944±0.011		0.004
-2.0V _{SHE}	0.014	Ti-O	4.8±0.7	1.946±0.015		0.003

These experimental results are further supported by constant-potential density functional theory (DFT) calculations. As the negative electrode potential becomes more negative from 0 V_{SHE} to -2.0 V_{SHE}, the negative surface charge of the TiO₂ (101) facet increases linearly, while the formation energies of oxygen vacancies decreases correspondingly, see Table R5. This accumulation of surface charge is directly linked to the formation of Ti³⁺ polarons, as clearly illustrated in Figure 1 of the main text. The formation of Ti³⁺ polarons, in turn, lowers the oxygen vacancy formation energies (Table R5) and indicates a strong correlation between the Ti³⁺ polaron generation and oxygen vacancy formation. Additionally, the computational results show that the average Ti-O bond lengths around the Ti³⁺ polaron site are elongated from 1.99 to 2.009 Å. The consistent trends observed in both theoretical and experimental studies confirm that the CIP-DFT method captures the atomic-scale structural changes under constant-potential conditions.

Table R5. The formation energies of an oxygen vacancy (E_{V_o}) on TiO₂(101) at PZC and different electrode potentials at the GGA+U level.

Electrode potentials (V _{SHE})	E_{V_o} (unit:eV)	Charge(e ⁻)
PZC	2.97	0
0	2.73	-0.26
-0.5	2.56	-0.67
-1.0	1.87	-1.04
-2	0.68	-1.81

Modifications: Table R4 can be found from page 22 in SI and Table R4 as table S9 on page 42 together with related text.

6. The EXAFS fitting lacks sufficient resolution to draw meaningful conclusions. The integral deviation in coordination numbers implies high uncertainty, and bond distance values with only two significant figures do not indicate any appreciable structural change during HER. The authors should provide a more robust fitting analysis with improved statistical confidence.

We have conducted a more robust fitting analysis, which shows that the uncertainty in the coordination number is now within $\pm 20\%$, and deviations in bond length are controlled to approximately ± 0.02 Å. These error margins fall within the acceptable range for the XAFS analysis ($\text{CN} \pm 20\%$; $R \pm 1\%$; $\sigma^2 \pm 20\%$). [Jing Zhang* et al. *Angew. Chem. Int. Ed.* 2024, 63, e202407509. <https://doi.org/10.1002/anie.202407509>]

In addition, standard TiO_2 and $\text{V}_\text{O}\text{-TiO}_2$ samples were subjected to the in-situ electrochemical spectroscopic analysis for comparative evaluation. The results show a slight decrease in the coordination number between Ti-O and a minor elongation in the Ti-O bonds in the $\text{V}_\text{O}\text{-TiO}_2$ system as the applied potential is made more reducing, see Table R6. It is important to note, that EXAFS measurements yield an "average" representation of bulk and surface contributions as discussed in point 4. However, due to the relatively small contribution of surface atoms to the overall signal, significant spectral changes are unlikely. Therefore, the CIP-DFT method employed in this study offers valuable atomic-scale insight into potential-dependent polaronic structural dynamics, effectively complementing the limitations of experimental, macroscopic averaged experimental data.

Table R6. EXAFS fitting parameters at the Ti K-edge.

Potential	R-factor	Path	C.N.	R (Å)	ΔE_0 (eV)	σ^2 (Å ²)
TiO_2	0.019	Ti-O	6*	1.928 \pm 0.012		0.005
$\text{V}_\text{O}\text{-TiO}_2$	0.014	Ti-O	5.2 \pm 0.5	1.934 \pm 0.021		0.005
OCP	0.013	Ti-O	5.2 \pm 0.6	1.942 \pm 0.010	-4.86	0.005
-1.0V _{SHE}	0.015	Ti-O	5.0 \pm 0.7	1.944 \pm 0.011		0.004
-2.0V _{SHE}	0.014	Ti-O	4.8 \pm 0.7	1.946 \pm 0.015		0.003

C.N., σ^2 , R-factor represented the coordination number, Debye-Waller factor, and goodness of fitness, respectively. R referred to the bond distance. The energy shift (ΔE_0) was set for all spectra to highlight the variation of R. S_0^2 was set to 0.68, according to the experimental EXAFS fitting of TiO_2 reference by fixing CN as the known crystallographic value.

7. The in-situ Raman analysis focuses narrowly on a single Eg peak, which predominantly reflects bulk TiO₂ properties. A thorough examination of all characteristic Raman modes (A_{1g}, B_{1g}, and E_g) is necessary to assess potential structural or electronic changes during HER. Additionally, the reported fluctuation of the E_g peak—attributed to varying oxygen vacancy concentrations—is not corroborated by XAS data, further weakening this interpretation.

We have now analyzed all characteristic Raman modes of pristine TiO₂ at different electrode potentials. As shown in Figure R7a, the application of increasingly negative potential leads to pronounced weakening and fluctuations of the E_g, B_{1g} and A_{1g} characteristic peaks. Among the E_g peak, the dominant feature of anatase TiO₂, arises from the symmetrical bending vibration of oxygen atoms within a specific crystal plane. The B_{1g} peak corresponds to the asymmetric stretching of oxygen atoms along a particular crystal direction. The A_{1g} peak is associated with the symmetric stretching of Ti-O bonds and reflects the degree of distortion of the octahedral [TiO₆] unit [*J. Phys. Chem. C* 2016, 120, 33, 18878-18886].

The intensity and frequency of the A_{1g} modes are sensitive to crystal defects, stress states and phase transitions. As shown in Figure R7c, a notable weakening of the Ti-O vibration intensity is observed in more negative potentials, suggesting increased distortion of the [TiO₆] octahedra or enhanced lattice disorder. These observations point to potential-induced lattice distortions, which we attribute to the formation of the Ti³⁺ polarons under negative bias.

Figure R7. (a) Raman spectra of TiO₂ at different electrode potentials. (b) The main E_g peak of TiO₂ at different electrode potentials. (c) The remaining three Raman peaks of TiO₂ at different electrode potentials.

Modifications: Figure R7 and above discussion has been added into the in Supporting Information on pages 46 and 47 in blue.

Reviewer 5.

In this revised manuscript, the authors have addressed several of the initial concerns; however, a critical issue raised: whether the spectroscopic changes observed during in-situ measurements are attributed to the formation of intermediates (e.g., polarons) under catalytic conditions or to irreversible alternations in the titanium oxidation state induced by the application of high negative potentials. This distinction is central to the authors' main claim—that polarons are generated in situ in a reversible manner. Given this, further clarification and supporting evidence are essential before the manuscript can be considered for publication.

We thank the reviewer for careful consideration of our work. Below we address in detail all raised issues and demonstrate that surface polarons form under the reaction conditions.

1. The reviewer appreciates the authors' explanation in the response letter regarding the time scale discrepancy between the experimental EPR measurements and theoretical CIP-DFT-MD simulations. However, the manuscript itself contains only a brief mention of this limitation. Since such divergence could mislead readers unfamiliar with the constraints of time-resolved simulations, a more detailed explanation – akin to what was provided in the response – should be incorporated into the main text.

We have improved the discussion on this matter, and the following text has been added in the manuscript.

Modification: *While CIP-DFT-MD provides the unique possibility of directly simulating potential-dependent polaron formation and dynamics in semiconductors,^{4,6} it should be noted that achieving the experimental (milli)second timescale of e.g. NAPXPS or EPR measurements is not possible with (CIP-)DFT, machine learning potential, or even classical MD simulations. Nevertheless, the CIP-DFT-MD simulations in Figures 2, 3, and S13 complement the experiments, and directly demonstrate that potential-driven Ti^{3+} surface polarons are both thermodynamically and dynamically stable at room temperature under reducing conditions on the picosecond timescale.*

2. The current in-situ Ti K-edge XAS measurements, which primarily probe bulk states, are insufficient to conclusively establish surface-localized Ti^{3+} formation during electrocatalysis. The authors did not conduct in-situ surface-sensitive techniques (e.g., Ti L-edge XAS or near-ambient pressure XPS), instead offering ex-situ Ti L-edge XAS results without detailed experimental description. This omission raises concerns: if any change is observed in ex-situ measurements, it could represent irreversible structural modification rather than the presence of reaction intermediates. Such changes might result from the application of high negative bias rather than the catalytic process itself, contradicting the manuscript's central claim of reversible in-situ polaron formation.

Thank you for your suggestion. Following your advice, we performed state-of-the-art, in-situ near-ambient pressure X-ray photoelectron spectroscopy measurements (NAPXPS) to investigate

the surface electronic and chemical states of TiO₂ as a function of applied electrode potentials. As shown in Figure R1c, applying a reducing potential of $U = -2.2 \text{ V}_{\text{SHE}}$ gives rise to new XPS peak at a lower binding energy, which we attribute to the formation of Ti³⁺ polarons on the TiO₂ surface. This results from the fact that in XPS, a lower binding energy for Ti³⁺ compared to Ti⁴⁺ reflects its reduced effective nuclear charge: the lower oxidation state weakens the Coulombic attraction between the nucleus and electrons, shifting the 2p peak to lower binding energy.

Importantly, the onset potentials for Ti³⁺ polaron formation derived from in-situ electrochemical EPR and in-situ near-ambient pressure XPS (Figure 2a and Figure R1c) show excellent agreement, differing by only 0.1 V_{SHE}. This consistency indicates that the potential-dependent formation of Ti³⁺ polarons primarily occurs at the surface of TiO₂. Furthermore, these experimental observations are strongly supported by our computational CIP-DFT results: the difference in the polaron formation potential among NAPXPS, EPR, and CIP-DFT is within 0.2 V vs SHE, confirming the robustness of our main finding – the potential-driven formation of polarons.

To further demonstrate that the polaron formation is reversible and not e.g. due to irreversible structural changes, we checked that the polaron signal in the NAPXPS disappears once the electrode potential is returned back to the open-circuit potential (OCP) where the polaron signal was initially absent: the results in Figure R1c show that the Ti³⁺ appearing at $U = -2.2 \text{ V}_{\text{SHE}}$ disappears as the potential is brought back to the OCP, confirming that the potential-driven formation of Ti³⁺ polarons is a reversible, in situ process.

Furthermore, we performed a series of control measurements using in-situ EFM, EPR, and Raman spectroscopies. The results show that the voltage-driven Ti³⁺ polarons disappear within 1 minute after removal of the applied voltage in the range of -1 V to -3 V, as shown in Figure R2b. Moreover, under higher voltages of -5 V and -7 V, the Ti³⁺ polaron disappears within 3 minutes after voltage removal, see Figure R2c and R2d. The absence of pronounced features in the EFM-2 ω signal suggests that the dielectric constant and thus the stoichiometry of the TiO₂ sample remains unchanged under different voltage conditions, see Figure R2e-h. In-situ EPR spectroscopies also demonstrate that the signal of potential-dependent Ti³⁺ polaron formation disappears under open-circuit conditions after removal of the electrode potential at $U = -3.0 \text{ V}_{\text{SHE}}$, see Figure R3a. In-situ Raman spectroscopies also reveal that the weakened E_g, B_{1g} and A_{1g} characteristic peaks in TiO₂, which are induced by potential-dependent Ti³⁺ polaron formation, return to their initial states under open-circuit conditions after removal of the electrode potential at $U = -3.0 \text{ V}_{\text{SHE}}$, see Figure R3b. Therefore, **these results support the manuscript's central claim regarding the in-situ and reversible formation of Ti³⁺ polarons at the TiO₂ surface.**

Figure R1. (a) An optical image of in situ electrochemical cell in NAPXPS chamber. (b) A schematic of the in situ electrochemical measurements. (CE=counter electrode, RE=reference electrode, WE=working electrode) (c) in-situ electrochemical NAPXPS measurements.

Figure R2. (a) EFM- 1ω observed on the pristine TiO_2 sample subjected to tip biases in a range of $-1 \sim -7\text{V}$ at different locations. EFM- 1ω observed on the pristine TiO_2 samples under open-circuit conditions at 1 min (b), 3 min (c), and 5 min (d) after voltage removal. (e) EFM- 2ω measurements on the pristine TiO_2 sample subjected to tip biases in a range of $-1 \sim -7\text{V}$ at different locations. EFM- 2ω observed on the pristine TiO_2 samples under open-circuit conditions at 2 min (f), 4 min (g), and 6 min (h) after voltage removal.

Figure R3. (a) In situ EPR spectroelectrochemical measurements of TiO_2 sample. (b) In situ spectroelectrochemical Raman measurements of TiO_2 sample.

Modifications: Figure R1 has been added into the manuscript on page 9, and the accompanying text is given in blue. Figure R2 and R3 have been added into the Supporting Information on pages 15, 16 and 46, and the related text is given in blue.

3. To align with the high standards expected by Nature Communications, the reviewer maintains that it is essential to perform in-situ Ti L-edge XAS or in-situ near-ambient pressure XPS to directly probe surface electronic and chemical states under operational conditions. These techniques are currently available and would provide critical evidence to support the manuscript's central claim regarding the in-situ and reversible formation of polarons. In addition, the authors are required to include control measurements under open-circuit conditions after voltage application. This is crucial to distinguish between reversible changes associated with catalytic intermediates and irreversible modifications induced by the application of high bias. Without such controls, it remains unclear whether the observed spectroscopic changes truly reflect catalytic dynamics or are artifacts of material degradation.

We have conducted the requested experiments. For a detailed discussion, please refer to the response provided in Comment 2.

4. The authors highlight a 0.23 eV shift in the Ti K-edge XANES region to infer valence state changes. However, considering the large energy difference between Ti^0 (~4965 eV) and Ti^{4+} (~4983 eV), a shift of this magnitude is negligible and corresponds to a minor oxidation state change (~0.05 e^-), which may not yield meaningful chemical insight. The authors are requested to perform quantitative linear fitting based on the edge inflection point to determine Ti oxidation states more accurately and re-evaluate whether such a minor energy shift can meaningfully support the claims of in-situ polaron formation.

The validity of the EXAFS fitting also warrants further scrutiny:

- The fitting models used for $Vo-TiO_2$ and pristine TiO_2 must be explicitly described, including structural parameters and assumptions.
- Experimental spectra and fitted curves should be presented together to demonstrate fit quality.
- High R-factors (approaching 0.02) diminish the reliability of subtle changes in coordination number and bond distances. More robust evidence is required to validate such interpretations.
- Notably, ΔE_0 values appear identical across TiO_2 and $Vo-TiO_2$ samples, which is inconsistent. ΔE_0 is a variable that reflects differences in electronic environment and should be independently optimized for each sample.

We thank the reviewer for these insightful comments. Upon H_2 plasma treatment, TiO_2 undergoes a distinct color change from white to dark gray, indicating the formation of abundant oxygen vacancies and Ti^{3+} polarons, as confirmed by the EPR spectra in Figure S30. The Ti K-edge absorption of $Vo-TiO_2$ is shifted by 0.46 eV relative to pristine TiO_2 , corresponding to an average valence change of approximately 0.1 e^- (Figure R4), reflecting a meaningful alteration in Ti's chemical state.

The fitting models for V_O-TiO₂ and pristine TiO₂ are now described explicitly in Table R1. Owing to the strong correlation between the energy shift (ΔE_0) and the scattering path length (R), and considering the similar valence and electronic structures of the two samples, ΔE_0 was fixed to a common value to ensure robust extraction of structural parameters (see *Nat. Catal.* **5**, 414–429 (2022); <https://doi.org/10.1038/s41929-022-00783-6>). The experimental spectra and fitted curves are shown in Figure R5.

Ti K-edge analyses of TiO₂ commonly exhibit relatively high R-factors (0.01–0.04) due to intrinsic limitations in data quality for TiO₂ samples (<https://pubs.acs.org/doi/10.1021/jp511739h>; <https://pubs.acs.org/doi/10.1021/acs.nanolett.1c01995>). Discussions with XAFS experts confirm that the quality of XAFS data for Ti is inherently poor. Our R-factors (0.01–0.02) fall within the range of high-quality fits that can be expected for TiO₂ but, as correctly noted by the reviewer, values approaching 0.02 reduce the sensitivity to subtle variations in coordination number and bond distance. **We have thus decided to exclude the XAFS data from the present work.**

Although the XAFS data have been removed, prior studies have clearly established that Ti–O bonds at Ti³⁺ sites can elongate to accommodate excess electronic density – a distortion essential for polaron localization (<https://pubs.acs.org/doi/full/10.1021/jp2001139>, <https://doi.org/10.1103/PhysRevLett.97.166803>, <https://doi.org/10.1021/acs.jpcc.9b05975>, <https://doi.org/10.1021/acs.jpcc.8b11501>). Our computational results are fully consistent with these findings, ensuring that the omission of XAFS data does not affect the validity of our conclusions.

Moreover, our *in situ* NAPXPS and *in situ* EPR measurements already provide direct and robust evidence for the reversible formation of Ti³⁺ polarons at the TiO₂ surface. Given the well-documented structural response of Ti³⁺ sites in literature, we excluded the XAFS data to maintain analytical reliability.

Figure R4. (a) Fitted average valence state of Fe from X-ray absorption near-edge structure (XANES) spectra.

Table R1. EXAFS fitting parameters at the Ti K-edge.

Potential	R-factor	Path	C.N.	R (Å)	ΔE_0 (eV)	σ^2 (Å ²)	R (Å)-DFT
TiO ₂	0.019	Ti-O	6*	1.928±0.012		0.005	
Vo-TiO ₂	0.014	Ti-O	5.2±0.5	1.934±0.021		0.005	
OCP	0.013	Ti-O	5.2±0.6	1.942±0.010	-4.86	0.005	1.991
-1.0V _{SHE}	0.015	Ti-O	5.0±0.7	1.944±0.011		0.004	2.003
-2.0V _{SHE}	0.014	Ti-O	4.8±0.7	1.946±0.015		0.003	2.009

C.N., σ^2 , R-factor represented the coordination number, Debye-Waller factor, and goodness of fitness, respectively. R is the average Ti-O bond distance. The energy shift (ΔE_0) was set for all spectra to highlight the variation of R. S_0^2 was set to 0.68, according to the experimental EXAFS fitting of TiO₂ reference by fixing CN as the known crystallographic value. The Ti-O distances are obtained from the DFT-optimized ground-state structure of anatase TiO₂, where anatase crystal phase is confirmed by experimental XRD results.

Figure R5. FT-EXAFS fitting curves.

5. From the authors' response and images provided, there are concerns about the suitability of the in-situ XAS setup:

- The explanation that bubble formation interferes with XAS measurements is questionable, as the in-situ cell is designed for back-side illumination, where the X-ray beam reaches the catalyst layer without passing through the electrolyte, minimizing bubble interference.
- The positioning of the catalyst relative to the Lytle detector appears incorrect. In standard setups, the sample should be positioned near the center of the fan-shaped detection grid (approximately one-third from the edge), regardless of left/right alignment. Misalignment here may compromise data integrity.

We sincerely thank the reviewers for their valuable comments on the in-situ XAS experiments. Based on the discussions in response to Comment 4, we have decided to remove the XAFS data from this study to ensure the reliability of the data.

Additionally, we also sincerely apologize for the lack of clarity in the labeling of the previous in-situ XAFS setup image. Due to the use of a photograph depicting the actual experimental arrangement, label placement may have introduced visual ambiguity. We have now revised the label to precisely indicate that the sample is positioned within the central region of the fan-shaped detection grid, consistent with the standard setups, see Figure R6b.

Figure R6. Digital images for in-situ XAFS test set-up.